# Turing structuring with multiple nanotwins to engineer efficient and stable catalysts for hydrogen evolution reaction

Jialun Gu[1,2,6,7,8], Lanxi Li[1,3,7,8], Youneng Xie [1,2,7], Bo Chen [4], Fubo Tian[5], Yanju Wang[1,2,7], Jing Zhong [3], Junda Shen[3,7] & Jian Lu [1,2,3,6,7] ✉

Low-dimensional nanocrystals with controllable defects or strain modifications are newly emerging active electrocatalysts for hydrogen-energy conversion and utilization; however, a crucial challenge remains in insufficient stability due to spontaneous structural degradation and strain relaxation. Here we report a Turing structuring strategy to activate and stabilize superthin metal nanosheets by incorporating high-density nanotwins. Turing configuration, realized by constrained orientation attachment of nanograins, yields intrinsically stable nanotwin network and straining effects, which synergistically reduce the energy barrier of water dissociation and optimize the hydrogen adsorption free energy for hydrogen evolution reaction. Turing PtNiNb nanocatalyst achieves 23.5 and 3.1 times increase in mass activity and stability index, respectively, compared against commercial 20% Pt/C. The Turing PtNiNb-based anion-exchange-membrane water electrolyser with a low Pt mass loading of 0.05 mg cm$^{-2}$ demonstrates at least 500 h stability at 1000 mA cm$^{-2}$, disclosing the stable catalysis. Besides, this new paradigm can be extended to Ir/Pd/Ag-based nanocatalysts, illustrating the universality of Turing-type catalysts.

High activity and stability are the two crucial motifs in electrocatalytic discipline[1]. Effective strategies for synthesizing highly active catalysts are constructing low-dimensional nanomaterials with strain or crystalline defect modulation. Lattice strain can optimize surface electronic structure by changing *d*-band center and bandwidth to tune adsorption energy of reaction intermediate on catalyst surface, and further improving catalytic activity[2–4]. Atom configuration of metallic catalyst surface is another pivotal factor to determine catalyst performance, especially surface terminations of crystalline defects, such as twins and stacking faults, which are usually active sites due to specific coordination structure and defect-induced strain[3,5]. However,

high surface energy and thermodynamic instability of strain/defects-powered low-dimensional nanocatalysts tend to induce strain relaxation, spontaneous surface reconstruction and transformation to twin-free Wulff constructions, thus resulting in poor structural and catalytic stability for long-term applications[6–9]. These limitations initiate the exploitation of new design guidelines for activating and stabilizing low-dimensional nanocatalysts.

The constructions of low-dimensional materials have mainly focused on structural controls for function realization, with few considerations on spatiotemporal controls[10]. Turing patterns, known as spatiotemporal stationary patterns, are widely observed in

[1]Centre for Advanced Structural Materials, City University of Hong Kong Shenzhen Research Institute, Greater Bay Joint Division, Shenyang National Laboratory for Materials Science, Shenzhen, China. [2]Department of Mechanical Engineering, City University of Hong Kong, Hong Kong, China. [3]Department of Materials Science and Engineering, City University of Hong Kong, Hong Kong, China. [4]Department of Chemistry, City University of Hong Kong, Hong Kong, China. [5]State Key Laboratory of Superhard Materials, College of Physics, Jilin University, Changchun, China. [6]CityU-Shenzhen Futian Research Institute, Shenzhen, China. [7]Hong Kong Branch of National Precious Metals Material Engineering Research Centre, City University of Hong Kong, Hong Kong, China. [8]These authors contributed equally: Jialun Gu, Lanxi Li. ✉e-mail: jianlu@cityu.edu.hk

far-from-equilibrium biological and chemical systems, such as the stripes of *Dania rero*, the regular surface coloring on sea-shells and the hexagonal arrays present in microemulsions[11–16]. The mehcanism of these pattern formations is related to the reaction-diffusion theory proposed by A.M. Turing, in which the activator with a smaller diffusion coefficient induces local preferential growth[15]. The common visualized shape of Turing patterns are hexagonally packed cylinders, spots and labyrinthine patterns[10,14]. These Turing patterns are spontaneous symmetry breaking in pristine homogeneous systems[17]. This topological characterization, emerging in nanoscale Turing patterns, might be implemented by the anisotropic growth of nanograins. Such broken lattice symmetry has crucial crystallographic implications for the growth of specific configurations, such as two-dimensional (2D) materials with intrinsic broken symmetry and twinning. Inspired by the relevance between crystalline symmetry and morphogenesis, Turing structuring may provide a new structure motif to engineer the growth of low-dimensional materials with strain and defect modifications. The phenomenological two anti-phases and fertile boundaries in Turing patterns are of high interest for interface dominated applications, particularly electrocatalysis. Therefore, it is scientifically significant to explore the application of Turing theory on nanocatalyst growth and the relations with crystallographic defecting.

Herein, we describe the fabrication of physical-vapor-deposited platinum–nickel–niobium (PtNiNb) nanosheets that exhibited a supra-nanometer-sized (<10 nm) Turing structure and thus functioned as an efficient electrocatalyst. The Turing stripes are formed from inter-confined nanograins with different orientations, and constrained orientation attachment during their formation resulted in the high-density nanotwins and large lattice strain. Turing structuring endow these Turing PtNiNb nanosheets with long-term stability and high mass activity in the alkaline hydrogen evolution reaction (HER), with these properties being more than one order of magnitude greater than those of the commercial Pt/C catalyst. Density functional theory (DFT) calculations demonstrate the synergistic effect of twin boundary and strain to accelerate water dissociation and optimize electronic structure as well as hydrogen adsorption free energy.

## Results

### Preparation and characterization of turing PtNiNb catalyst

Superthin PtNiNb nanosheets were prepared by a two-step sputtering approach, sputtering a silicon precursor layer followed by co-sputtering of a metal layer. The super thin film was exfoliated from the substrate and subsequently ultrasonicated to form the nanosheet catalyst ink (see Methods). Transmission electron microscopy (TEM) images reveal that the catalysts consisted of free-standing PtNiNb nanosheets with lateral sizes ranging from 100 to 500 nm (Fig. 1a and Supplementary Fig. 1). The thickness of PtNiNb nanosheet could be tuned by the newly developed method in the range from 4 nm to

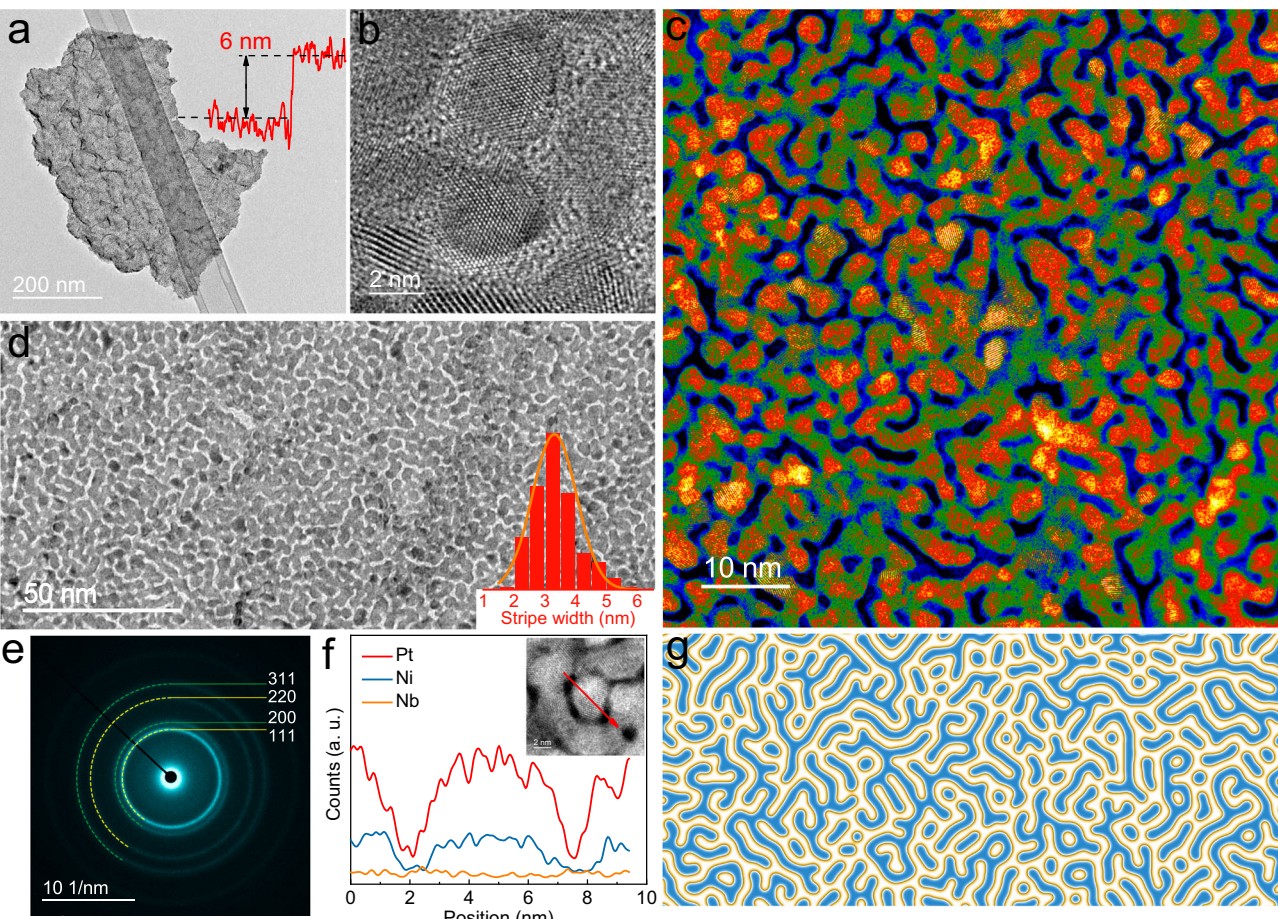

**Fig. 1 | Structure and morphological characterization of Turing PtNiNb. a** Low-magnification TEM image of free-standing Turing PtNiNb with a thickness of 6 nm. The inset is the height profile across the edge of Turing PtNiNb. **b, c** High-resolution TEM and HAADF-STEM images showing Turing-type structures, respectively. The Turing stripes consisted of nanograins that met at the Y-type bifurcations. **d** TEM image of the uniformly distributed Turing stripes. The inset is the size distribution of Turing stripes in terms of the diameter of constituent nanograins. **e** SAED pattern from **c**, indexed with a face-centered cubic structure. **f** The STEM-EDS line-scanning analysis of a Turing stripe. The inset shows the analyzed stripes and the red arrow represents the line-scanning direction. **g** Schematic diagram of typical Turing structure.

12 nm. This is thinner than certain 2D metals synthesized by organic ligand-confined growth and liquid-phase exfoliation[18,19].

The high-angle annular dark-field scanning TEM (HAADF–STEM) images of the PtNiNb nanosheets reveal their striped patterns with the characterization of two phases and clear domain boundaries (Fig. 1b, c). The irregular stripes were connected to each other to form labyrinthine networks. Higher-magnification images show that the stripes were composed of individual supra-nanometer-sized grains with various orientations (Fig. 1b, c and Supplementary Fig. 2). Another topological feature of these PtNiNb nanosheets are the highly branched domains, i.e., Y-shaped bifurcations, which formed where stripes facing in different directions met. Labyrinthine pattern composed of irregular stripes in the superthin PtNiNb nanosheets topologically resembles Turing patterns that are ubiquitous in biological systems (e.g., stripes on *Dania rero* and on the surface of seashells) and chemical systems (e.g., the Belousov–Zhabotinsky reaction and the chlorite–iodide–malonic acid reaction, Fig. 1g)[12–14]. These patterns are studied as morphogenesis models in Turing's theory. Therefore, we denote these PtNiNb nanosheets as Turing PtNiNb in this work due to their Turing-type topological features.

The low-magnification TEM image shows that Turing stripe networks were uniformly distributed throughout Turing PtNiNb, with a size distribution ranging from 2 to 6 nm and an average size of 3.3 nm (Fig. 1d). Turing PtNiNb is one order of magnitude smaller than the most recently reported nanoscale chemical Turing patterns, and significantly smaller than the macroscopic/microscopic Turing structures found in biological systems[11,20].

The selected area electron diffraction (SAED) patterns of Turing PtNiNb display the sharp diffraction rings of a face-centered cubic (fcc) structure (Fig. 1e). This indicates that the stripes of Turing PtNiNb comprised nanograins with random orientations. The powder X-ray diffraction patterns confirm the fcc structures of 2D monometallic Pt and Turing PtNiNb. (Supplementary Fig. 2c). The composition of Turing PtNiNb was $Pt_{56.1}Ni_{33.5}Nb_{10.4}$, determined by inductively coupled plasma–optical emission spectroscopy (ICP-OES). The line-scanning and mapping of STEM-energy-dispersive spectroscopy (STEM-EDS) demonstrate the homogeneous distribution of Pt/Ni/Nb in the stripes and gaps (Fig. 1f and Supplementary Fig. 3). The gaps between Turing stripes (the low-contrast areas in the HAADF-STEM and TEM images) had a similar composition but a much lower elemental signal intensity than the stripes themselves, suggesting that the thickness of gaps is smaller than the latter. Therefore, the Turing stripes on Turing PtNiNb were quasi-three-dimensional structures rather than planar patterns. Turing PtNiNb exposed both top, bottom surfaces, and two side surfaces, and thus had a much larger specific surface area than that of typical planar 2D metals. The unique surface configuration of Turing PtNiNb, which hosts abundant exposed crystal facets and interfaces, is a distinct advantage for catalytic application.

A series of Pt-Ni-Nb nanosheets were further synthesized to determine the composition-structure relations in Turing catalyst. The Pt content in these new nanocatalysts was tailored in the range from 44.3 at% to 76.9 at%, and the ratio of nickel to niobium is fixed at approximately 3:1 (the same with the Turing PtNiNb). As the platinum content decreasing from 56.1 at% to 51.7 at%, the stripes tend to grow in-plane and parts of coarsening stripes interconnected to form large junctions (Supplementary Fig. 4c). Furthermore, the traits of Turing structure eventually disappeared in $Pt_{44.3}Ni_{42.1}Nb_{13.6}$ sample (Supplementary Fig. 4a, b). However, higher platinum contents could also cause the breakdown of Turing structure. As shown in Supplementary Fig. 4, $Pt_{60.1}Ni_{30.8}Nb_{9.1}$, $Pt_{69.2}Ni_{23.6}Nb_{7.2}$, $Pt_{73.4}Ni_{20.1}Nb_{6.5}$, and $Pt_{76.9}Ni_{17.8}Nb_{5.3}$ samples are planar nanosheets without the features of Turing structures. All the high-platinum-content samples ($\geq 60.1$ at%) are composed of classical nanocrystals and do not have particular topographical characteristics. These results suggest that the composition of Turing structure seems to be limited to a narrow range for

Pt-Ni-Nb alloys. In addition, we synthesized the binary $Pt_{63.7}Ni_{36.3}$ nanosheet (denoted as PtNi 2D metal) that has a Pt-Ni ratio of 1.75 close to the ratio of Pt to Ni (1.67) in Turing PtNiNb ($Pt_{56.1}Ni_{33.5}Nb_{10.4}$). TEM images of the $Pt_{63.7}Ni_{36.3}$ catalyst show the nanocrystals with random orientations (Supplementary Fig. 5). The nanosized crystals are densely arranged with clear grain boundaries. This kind of classical morphology of nanometallic materials is similar to the microstructures of the high-platinum-content Pt-Ni-Nb samples discussed above and is in sharp contrast to Turing structure. The niobium-deletion in Pt-Ni alloys would lead to the collapse of Turing structure and this result underlines the important contribution of niobium doping to induce the formation of Turing structure.

High-resolution lattice images were obtained for detailed crystallographic characterization (Fig. 2). Numerous twins and crystalline defects were observed in the nanograins, including five-fold, two-fold twins, stacking faults and lattice distortions. The frequency ratio of five-fold, two-fold and typical twins was 55%:18%:27%. Many bifurcations were made up of five-fold twins enclosed by {111} and {200} atomic planes (Fig. 2a, b). The fast Fourier transform (FFT) patterns of the five-fold twins were the typical diffraction spots of {111} and {200} planes that were imaged along the [110] zone axis (Supplementary Fig. 6). The atomic arrangement on both sides of the {111} twin boundary (TB) was fully symmetric and had the stacking sequence ABC|CBA. This configuration corresponds to a {111} coherent twin boundary (Σ3 {111} TB, CTB)[21]. In the fcc structure, the theoretical angle between {111} planes are 70.35° (Fig. 2d). The space-filling of such a five-fold symmetric structure requires compensation of a 7.35° misfit angle. In nanoparticles and nanoprecipitates with an fcc structure, the 7.35° misfit is typically compensated by a wedge disclination along the [110] direction for {111} TBs[22,23]. In the current study, edge disclination was not observed in the atomic-resolution images (Fig. 2a, b). However, the angles between the Σ3 {111} TBs in the five-fold twins were different and deviated from the theoretical value (Fig. 2a, c). These inhomogeneous TB angles indicate the heterogeneous distribution of the misfit within the twins, leading to large lattice strains that were further confirmed by the difference in lattice constants (Fig. 2e).

We used geometric phase analysis to quantitatively describe the strain distribution in these Turing PtNiNb[24]. Strain mapping reveals the heterogeneous distribution of lattice strain related to the TB intersection angles (Fig. 2f). Most parts of the twin from $Σ3_3$ to $Σ3_4$ were expanded due to tensile strain, the average lattice strain was 4.3%. Tensile strain was also present in the twin from $Σ3_1$ to $Σ3_5$, which had an average lattice strain of 3.1%. In contrast, the twin from $Σ3_4$ to $Σ3_5$ was under compressive strain and thus had an average lattice strain of 2.6%. The lattice strain was distributed inhomogeneously in the twin lattices, and the maximum strains occurred at edges and TBs rather than the dipole of five-fold twins or the center of the subunits[22,23]. The strain distribution of five-fold twins in Turing PtNiNb is thus significantly different from that of the five-fold twinned nanoparticles with a free boundary[25]. In such unconfined nanoparticles, crystals rotate randomly according to the accommodation of orientation attachment[26], and thus their maximum lattice strain occurs at the dipoles of five-fold twins[22]. The heterogeneous distribution of lattice strain evidences that the multiply twinning is a crystallographic protocol of the Turing structure to accommodate the orientation misfit of inter-constrained nanograins at bifurcations. Although twinning greatly alleviated orientation misfit, existing crystals could not be fully reconstructed for orientation accommodation owing to the confinement from the substrate and neighboring grains. Therefore, inhomogeneous-distributed residual lattice strain was preserved in the inter-constrained nanograins and thus in the structure of Turing PtNiNb. Orientation accommodation in this Turing structure was also realized by twinning–twinning and twinning-stacking faults (Fig. 2g and Supplementary Fig. 7). In addition, residual lattice misfit and the lattice distortion originating from stacking faults induced large strain in the

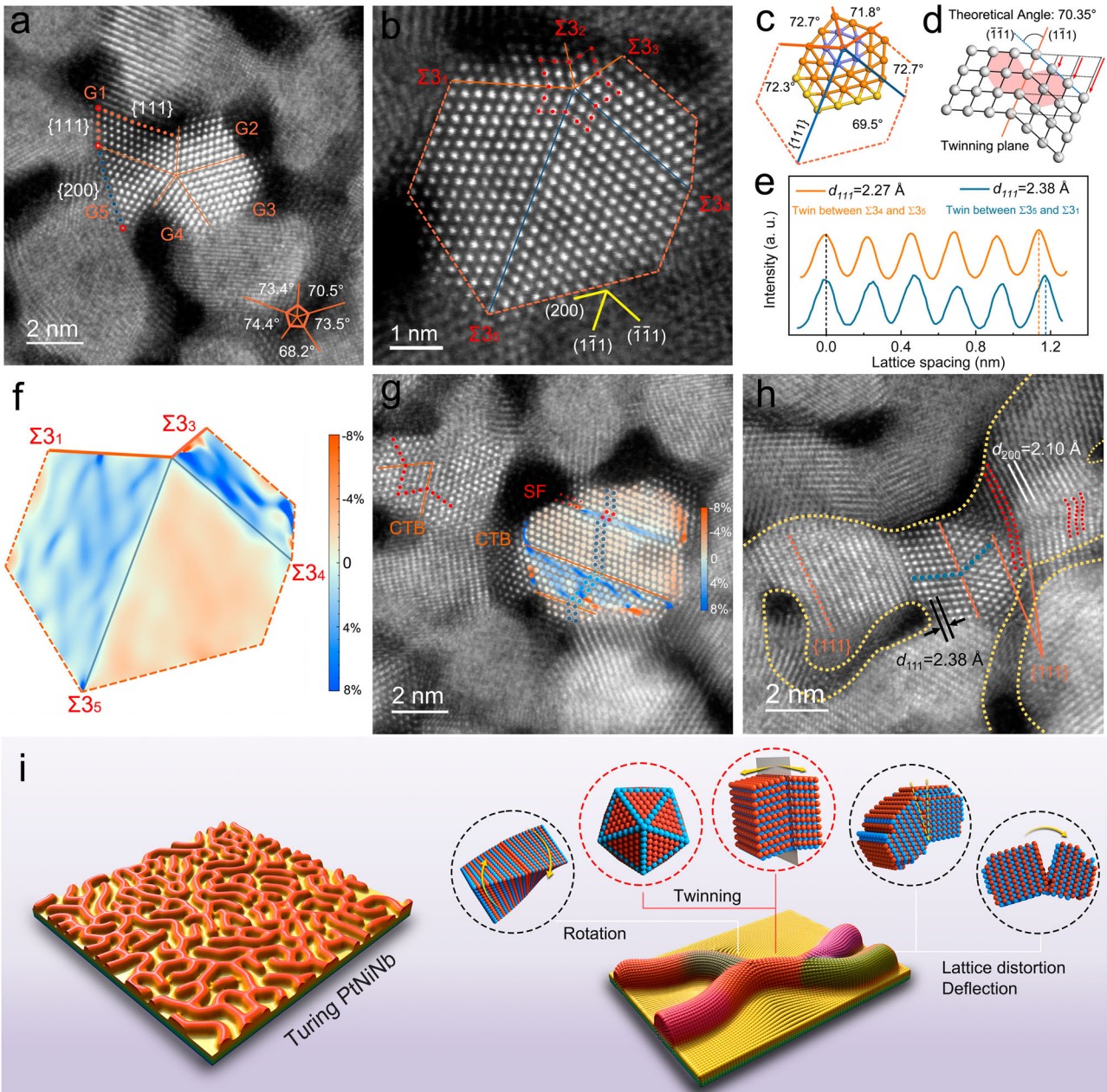

**Fig. 2 | Microstructures of Turing stripes with high-density defects and lattice strain. a** Atomic-resolution HAADF-STEM image of a five-fold-twinned bifurcation in a Turing stripe. The five-fold twins unify the neighboring nanograins with different orientations. G, grain. **b** Atomic arrangement at the bifurcation of Turing stripes. The orange and green solid lines denote the Σ3{111} CTBs. **c** The angles between the Σ3{111} twin boundaries in the five-fold twins shown in **b**. **d** The schematic diagram showing the theoretical angle between (_1_11) and (1_11) in the fcc structure. **e** The intensity profiles along {111} facets of the sub-units in the five-fold twins shown in **b**. **f** Strain mapping of the dashed-line marked area of the five-fold twins shown in **b**. **g** Constituent grains of the Turing stripes containing twins and stacking faults with the corresponding strain mapping. CTB, coherent twin boundary. SF, stacking fault. **h** Illustration of a Turing stripe formed by end-to-end-jointed grains oriented in a similar axial direction, that was accommodated by rotation/torsion of nanocrystals, twinning and lattice distortion. **i** Schematic diagram of the prepared Turing PtNiNb and corresponding crystallographic characterization.

grains (Fig. 2g). Consequently, the Turing structure of Turing PtNiNb is formed through constrained orientation attachment of sub-ten-nanometer-sized grains. Figure 2h shows a Turing stripe comprising a rotational crystal (rotated alone <111 > ), twins, a deflected crystal and local regions with large lattice distortions (marked by red spots). The high-density multiple twins and large lattice strain of these Turing stripes originated from their unique crystalline construction, and were thus intrinsic features of Turing PtNiNb.

The Turing PtNiNb presents a class of novel topological structure in nanometallic materials. We performed X-ray absorption near-edge

structure (XANES) and extended X-ray absorption fine structure (EXAFS) experiments to reveal the electronic structure and coordination information of Turing structure. The Pt L$_3$-edge XANES spectra show that the absorption thresholds and white line peaks of the tested samples Turing PtNiNb (Pt$_{56.1}$Ni$_{33.5}$Nb$_{10.4}$), Pt$_{56.1}$Ni$_{33.5}$Nb$_{10.4}$ (PtNiNb, without Turing structure), bimetal Pt$_{63.7}$Ni$_{36.3}$ and Pt 2D metal are close to those of the Pt foil, suggesting the Pt valence states are almost identical to zero (Fig. 3a)[27]. The Fourier-transformed EXAFS spectra of Pt foil and Pt 2D metal shows the similar coordination environment, and the corresponding dominating peaks are assigned to Pt-Pt

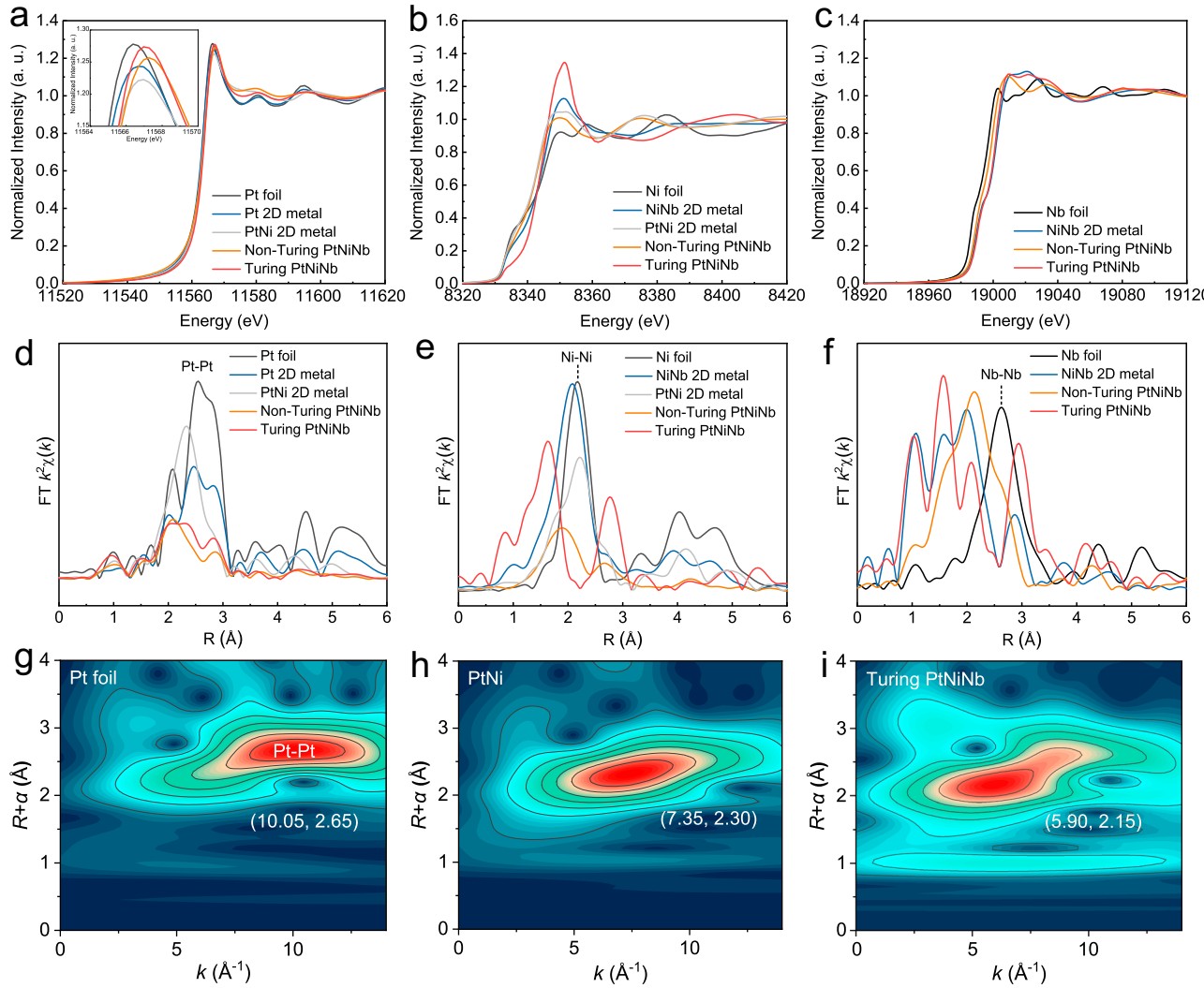

**Fig. 3 | Electronic structure analysis.** Electronic structure analysis. XANES spectra collected at Pt L₃-edge **a**, Ni K-edge **b**, and Nb K-edge **c**. Fourier transformed EXAFS spectra at Pt L₃-edge **d**, Ni K-edge **e**, and Nb K-edge **f**. Wavelet transforms analysis of the Pt L₃-edge for Pt foil **g**, PtNi 2D metal **h**, and Turing PtNiNb **i**.

(~ 2.77 Å) coordination (Fig. 3d). Ni doping results in the left shift of the dominating peak in the EXAFS spectra of binary PtNi sample, which is resolved as the Pt-Ni (2.62 Å) and Pt-Pt (2.69 Å) coordination. In contrast, Turing PtNiNb and PtNiNb have greatly different EXAFS spectra with significantly reduced amplitude of the atomic shell peaks. The fitting results of EXAFS spectrum reveal the major coordination of Pt-Pt (2.72 Å), Pt-Ni (2.59 Å) and Pt-Nb (2.67 Å) in Turing PtNiNb sample. The changes in peaks amplitude suggest the reduced coordination number. The fitting results show that the coordination number values of both Turing PtNiNb (8.1) and PtNiNb (7.4) are lower than those of bimetal PtNi (9.5), Pt 2D metal (9.5) as well as Pt foil (12, Supplementary Table 1). Such pronounced reduction in coordination number is attributed to the small nanosheet size/thickness, fertile edge structures and substantial crystalline defects, including the nanotwins and stacking faults. The Pt coordination environment was further analyzed by wavelet transform (WT) spectra at Pt L3-edge. The WT spectrum in Fig. 3g shows that the maximum peak intensity of the Pt foil is located at 10.05 Å⁻¹, which originates from the Pt-Pt bond. The maximum peak intensity of bimetal PtNi shifts to 7.35 Å⁻¹ due to the contribution of Pt-Ni bond (Fig. 3h). Furthermore, the Pt-M paths of Turing PtNiNb shift to a lower k space, in sharp contrast to the Pt-Pt coordination in Pt foil (Fig. 3i).

Figure 3b shows the normalized Ni K-edge XANES spectrum. A clear pre-edge around 8333 eV was detected, suggesting the dipole-forbidden, quadrupole-allowed transition (1$s$ to 3$d$)[28]. The absorption edge of the XANES spectrum was applied to evaluate the Ni valence. Turing PtNiNb presents a higher adsorption threshold and a greater intensity of white line compared with the Ni foil, indicating the variations in electronic structure. Ni in Turing PtNiNb is speculated to be positively charged and parts of nickel species are oxidation states. The Ni valence states were further confirmed with the Ni 2$p$ high-resolution X-ray photoemission spectra (XPS) showing the characteristic peaks of Ni²⁺ (Supplementary Fig. 8). The Fourier transformed EXAFS spectra of the bimetal PtNi sample show a dominating peak at around 2.2 Å, which originates from the first shell of Ni-Ni coordination (Fig. 3e). The coordination environments of bimetal PtNi highly resemble those of Ni foil and NiNb nanosheet. Taking the bimetal PtNi as the reference, Nb doping contributes to the left shift in the dominating peaks of PtNiNb. Fitting results on the EXAFS spectrum of PtNiNb illustrate that the major coordination is Ni-Ni and the second peak located around 2.65 Å can be attributed to the Ni-Nb coordination. By contrast, Turing PtNiNb presents two distinct scattering paths in the EXAFS spectrum. The first peak at ~1.6 Å is assigned to Ni-O coordination, while the second peak at ~2.8 Å is speculated to originate from the second

coordination shell of the Ni-Ni scattering path. The Ni-Pt coordination with a small coordination number of 0.6 was also resolved from the fitting of EXAFS spectrum (Supplementary Table 2).

We also analyzed the XANES and EXAFS spectra of these samples at Nb K-edge. Turing PtNiNb, PtNiNb (without Turing structure), and NiNb samples show higher near-edge absorption energies and larger intensity of white line peaks than those of Nb foil, suggesting the higher Nb valence states in these three samples (Fig. 3c). The Fourier transformed EXAFS spectra of Turing PtNiNb and NiNb depicts the peak components between 1.2 Å and 1.9 Å as a result of Nb-O scattering pair, while this peak is relatively weak in PtNiNb sample (Fig. 3f). Furthermore, fitting calculations on the EXAFS spectrum of Turing PtNiNb resolved the Nb-Nb, Nb-Pt, Nb-Ni as well as Nb-O coordination. Turing structuring induced a significant reduction in coordination numbers. The coordination number of Nb in Turing PtNiNb (4.6) is even smaller than that of amorphous NiNb nanosheet (7.0) which represents a kind of low-coordination structure (Supplementary Table 3).

The greatly reduced coordination number of Ni and Nb elements in Turing PtNiNb indicate that the edge structures of Turing configuration, including the surface of the irregular stripes, interfaces, and twin boundaries, have abundant low-coordinated atoms (Supplementary Table 2 and 3). These findings are also consistent with the features of stacking-fault-activated Ag nanoparticle reported previously, in which stacking fault leads to a low coordination number[3]. In addition, the reduced coordination number could be attributed to the significantly increased proportion of surface atoms on the Turing stripes (the coordination number of surface atoms is generally lower than that of the inner atoms). This conclusion can be further supported by the greatly increased electrochemical active surface area (Supplementary Table 6) of Turing PtNiNb.

## Electrocatalytic properties and membrane-electrode-assembly performance

The low-dimensional noble metals with fertile crystalline defects are promising catalysts for diverse electrolysis, such as HER and ethanol oxidation reaction (EOR). The HER performance of Turing PtNiNb was first studied in a three-electrode system with an optimal Pt mass loading of 0.0628 mg cm$^{-2}$. Commercial Pt/C (20 wt%, with the Pt mass loading of 0.065 mg cm$^{-2}$), bimetal PtNi 2D metal, monometallic Pt 2D metal and amorphous NiNb 2D metal were tested under the same condition for comparison (their microstructures are shown in Supplementary Fig. 9). The linear sweep voltammetry (LSV) curves (Fig. 4a) show that Turing PtNiNb presents a much higher catalytic activity than these reference samples. The overpotential of Turing PtNiNb is only 18.0 mV to deliver a current density of 10 mA cm$^{-2}$ ($\eta_{10}$), while the $\eta_{10}$ values of bimetal PtNi 2D metal, Pt 2D metal, Pt/C and NiNb 2D metal are 27.2 mV, 33.6 mV, 51.0 mV, and 288.7 mV, respectively. Meanwhile, the mass activity, an important parameter to evaluate the catalytic activity, was calculated from the LSV curves normalized by Pt mass. Impressively, Turing PtNiNb achieved a considerably large mass activity of 7166.3 mA mg$^{-1}_{Pt}$ at 100 mV, which was 13.8 times that of Pt/C (517.9 mA mg$^{-1}_{Pt}$) and 6.8 folds higher than the monometallic Pt 2D metal (Fig. 4b). The Tafel slopes of Turing PtNiNb, bimetal PtNi 2D metal, Pt 2D metal, and Pt/C were all close to 29.9 mV dec$^{-1}$ (Supplementary Fig. 10), illustrating that all three species catalyze the HER via a Volmer-Tafel mechanism in which the Tafel step (2M-H$^*$ ⇌ H$_2$ + 2 M) is the rate-determining step[29]. The LSV curves without resistance compensation show similar activity tendency among Turing PtNiNb and other contrastive samples (Supplementary Fig. 11, electrolyser resistances can be found in Supplementary Table 4).

The catalytic performance of Turing PtNiNb was further estimated by glassy carbon electrode (GCE). LSV curves measured on GCE show Turing PtNiNb possesses the best performance with the lowest overpotential at 10 mA cm$^{-2}$ of 23.7 mV, which is lower than the control samples (PtNi 2D metal: 34.7 mV, Pt 2D metal: 44.5 mV, Pt/C: 53.4 mV

and NiNb 2D metal: 498.9 mV, Supplementary Fig. 12a). Moreover, Supplementary Fig. 12d. illustrates that Turing PtNiNb has a greatly higher mass activity of 3581.8 mA mg$^{-1}_{pt}$ with comparison of PtNi 2D metal (961.0 mA mg$^{-1}_{pt}$), Pt 2D metal (953.9 mA mg$^{-1}_{pt}$) and Pt/C (231.7 mA mg$^{-1}_{pt}$). Tafel slopes of Turing PtNiNb, PtNi 2D metal, Pt 2D metal and Pt/C were all close to 29.9 mV dec$^{-1}$ (Supplementary Fig. 12c, electrolyser resistances could be found in Supplementary Table 4). These findings are consistent with the previous electrocatalytic results conducted on the carbon cloth electrodes. In addition, the high catalytic performance of Turing was further analyzed by electrochemical impedance spectroscopy. The Nyquist diagrams show the minimum magnitude of the semi-circle of Turing PtNiNb among all the tested samples (Supplementary Fig. 13). Fitting results based on the equivalent circuit reveal that the charge transfer resistance of Turing PtNiNb (5.4 ohm) is lower than those of bimetal PtNi (6.8 ohm), Pt 2D metal (28.2 ohm) and NiNb nanosheet (46.5 ohm, solution resistances are listed in Supplementary Table 5), which could significantly enhance the reaction kinetics and support the excellent HER activity of the Turing PtNiNb.

We further calculated the double-layer capacitance ($C_{dl}$) and turnover frequency (TOF) to evaluate the quantity and H$_2$-generation ability of catalytic active sites. Turing structuring achieved a nearly three times increase in $C_{dl}$ value for the Turing PtNiNb (52.2 mF cm$^{-2}$) compared to the Pt 2D metal (17.5 mF cm$^{-2}$) and PtNi 2D metal (14.5 mF cm$^{-2}$) (Supplementary Fig. 14). The resulting large $C_{dl}$ of Turing PtNiNb demonstrates that the unique Turing structural configuration and the symbiotic high-density nanotwins contribute to the considerably increased active surface area for the low-dimensional nanosheet. Meanwhile. Turing PtNiNb exhibited a much larger TOF (27.5 H$_2$ s$^{-1}$) at the overpotential of 100 mV in the alkaline HER compared against Pt 2D metal (9.8 H$_2$ s$^{-1}$) and PtNi 2D metal (16.2 H$_2$ s$^{-1}$). (Supplementary Fig. 15). Intrinsic activity of catalysts is typically tuned by changing coordination structure or introducing lattice strain. The combination of nanotwins and lattice strain by Turing structuring achieves a clear leap in the intrinsic activity of low-dimensional nanosheet, and this value of Turing PtNiNb is greater than those of state-of-the art HER catalysts (Fig. 4c).

Despite the fact that many 2D metals with high electrocatalytic activity have been synthesized, the poor stability greatly hampers their development and application[6,7]. The long-term stability of Turing PtNiNb was evaluated by a chromo-potentiometric test at a large current density of 200 mA cm$^{-2}$ (Fig. 4d) Turing PtNiNb shows remarkable stability with a negligible potential difference of 5 mV even after 60 h operation. By contrast, the harsh stability test conditions resulted in pronounced overpotential changes of 170.5 mV, 187 mV, and 205.6 mV at the first 24 h for bimetal PtNi 2D metal, Pt 2D metal and Pt/C catalysts, respectively. The high catalytic stability of Turing PtNiNb was further evaluated comprehensively through potentiostatic measurement that revealed a near-constant current density (at −1.221 V, Supplementary Fig. 16c) and an accelerated durability test (Supplementary Fig. 16d). As illustrated in Fig. 4e, Turing PtNiNb exhibits excellent catalytic stability with no reduction in current density (at 60 mV) after 30,000 cyclic voltammetry (CV) cycles, whereas the current densities of bimetal PtNi, Pt 2D metal and Pt/C catalysts declined to 60.7%, 70.5%, and 65.9% of their initial values, respectively. In addition to the alkaline HER, Turing PtNiNb exhibited an excellent HER catalytic stability in 0.5 M H$_2$SO$_4$ solution and a high mass activity which is 24.8 times greater than that of commercial Pt/C catalyst (Supplementary Fig. 17a−d).

Then, we calculated the mass-normalized charge for H$_2$ production in stability measurements to compare the capability of continuous hydrogen generation, evaluating the potentiality for industrial application (calculation details see Methods). As shown in Fig. 4f, the excellent catalytic stability and activity of Turing PtNiNb contributed to an high mass-normalized charge for H$_2$ evolution (53.04 C g$^{-1}_{PGM}$) in

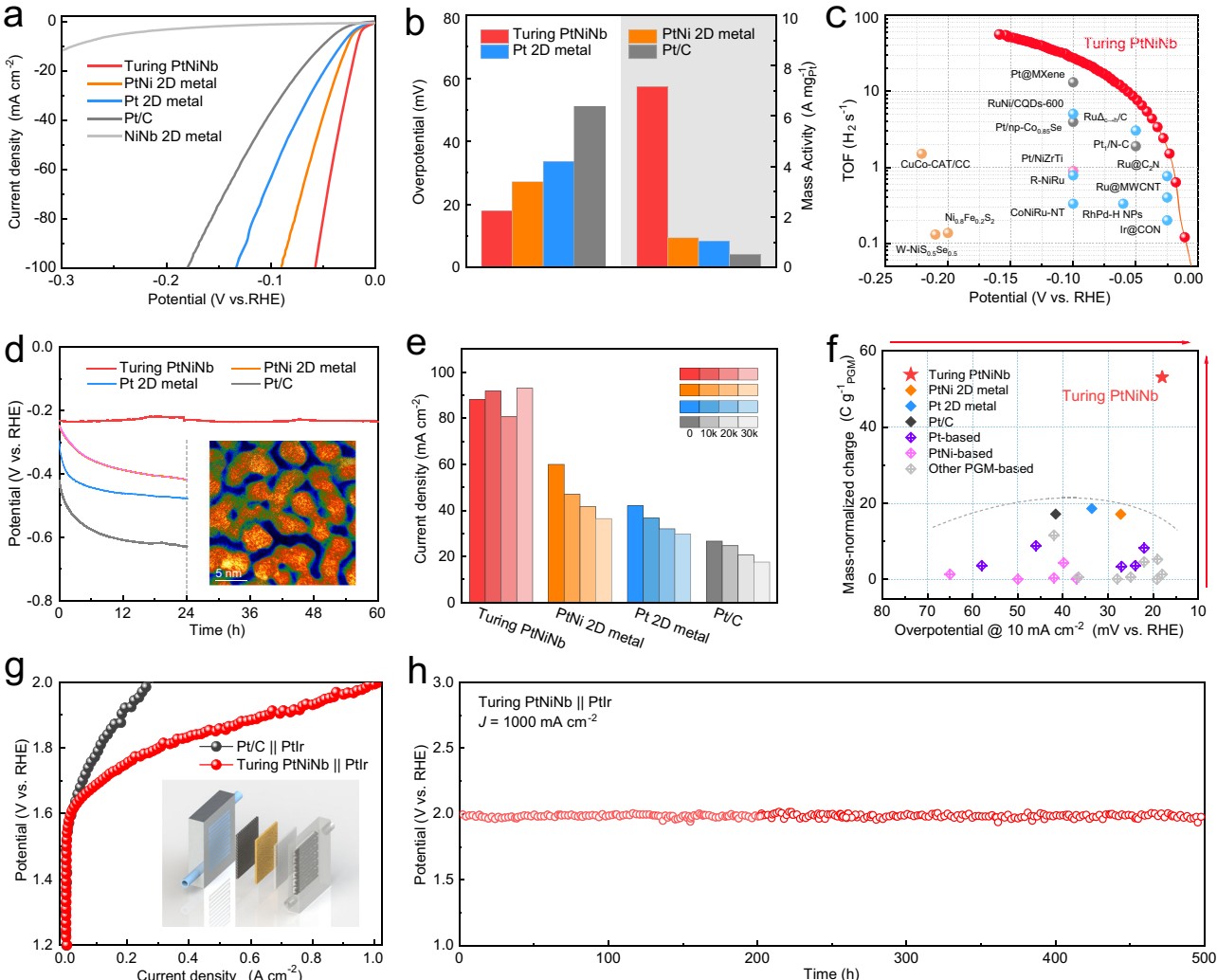

**Fig. 4 | Electrochemical performance of Turing PtNiNb. a** LSV curves of Turing PtNiNb, bimetal PtNi 2D metal, monometallic Pt 2D metal, commercial platinum and amorphous NiNb 2D metal in 1.0 M KOH at scan rate of 5 mV s⁻¹ with 85% resistance compensation. Resistances (ohm): 2.5 ± 0.03 (Truing PtNiNb), 3.1 ± 0.03 (PtNi 2D metal), 2.9 ± 0.05 (Pt 2D metal), 2.7 ± 0.03 (Pt/C) and 2.9 ± 0.07 (NiNb 2D metal). RHE, reversible hydrogen electrode. **b** Comparison for overpotential at 10 mA cm⁻² and mass activity at 100 mV (vs. RHE) between samples in **a**. **c** TOF of Turing PtNiNb and other recently reported electrocatalysts for alkaline HER. **d** Long-term stability test of Turing PtNiNb, PtNi 2D metal, Pt 2D metal and Pt/C under a large current density of 200 mA cm⁻² in 1.0 M KOH. The inset is the HAADF-STEM image of Turing PtNiNb after the stability test. **e** Variation of current density during accelerated durability test in 1.0 M KOH at scan rate of 200 mV s⁻¹. Columns refer to values at 60 mV recorded in LSV curves prior to testing, and after 10,000

(10k), 20,000 (20k), and 30,000 (30k) cycles. **f** Comparison of the mass normalized charge for hydrogen production in stability test and overpotentials at 10 mA cm⁻², with those of other reported HER electrocatalysts. PGM, platinum-group metal. The direction of the red arrows points to the superior performance. **g** Polarization curves of the anion-exchange-membrane water electrolysers using Turing PtNiNb or commercial Pt/C as the cathodic catalysts and PtIr gauze as the anodic catalyst in 1.0 M KOH solution. The inset is the schematic diagram of the anion-exchange-membrane water electrolyser comprising of the cathode polar plate, gas diffusion layers (carbon fiber paper), a membrane electrode assembly, a PtIr gauze and the anode polar plate from the left to right. **h** Chronopotentiometry tests of the Turing PtNiNb-based anion-exchange-membrane water electrolysers operated at 1000 mA cm⁻². All the electrochemical measurements were conducted under the room temperature.

the 60-h galvanostatic measurement, which is 3.1 times, 2.9 times and 3.1 times to those of bimetal PtNi (17.14 C g⁻¹_PGM), Pt 2D metal (18.60 C g⁻¹_PGM) and Pt/C catalysts (17.15 C g⁻¹_PGM), respectively. More importantly, the comprehensive catalytic property of Turing PtNiNb outperforms most literature reported nanocatalysts for alkaline HER tested in the experimental cell configurations. Thus, compared with other state-of-the-art electrocatalysts, Turing PtNiNb with superior capability to persistently and efficiently catalyze the H₂ generation shows greater potentiality in industrial anion-exchange-membrane (AEM) electrolysers.

The microstructure and composition of Turing PtNiNb (the sample shown in Fig. 4d) were investigated after the stability test. The Turing patterns indexed with an fcc structure were retained and the

average size of the Turing stripes was 3.3 nm, the same as the pristine sample (the inset of Fig. 4d and Supplementary Fig. 18). High-density five-fold twins and stacking faults were found in the HAADF-STEM images without any trace of structure reconstruction and degradation (Supplementary Fig. 18). The strain mapping near a stacking fault revealed that large lattice strain was still preserved, rather than strain relaxation, in the Turing structures after the long-term stability test (Supplementary Fig. 19). ICP-OES results showed that the composition change of the tested Turing PtNiNb was negligible (Pt₅₆.₀Ni₃₄.₄Nb₉.₆). The elemental valences and peak positions analyzed in X-ray photo-electron spectroscopy (XPS) spectra maintained the original status (Supplementary Fig. 20). The above findings are robust evidence of the structural and catalytic stability of Turing PtNiNb, indicating its high

resistance to composition and structure degradation driven by the electrical and chemical perturbations.

The implementation of Turing structure is an energetically favorable configuration for the quasi-3D PtNiNb nanosheets. CTBs networks are intrinsically stable and can stabilize the nanostructures[30]. High-density CTBs and nanograins were inter-constrained on Turing stripes—an efficient configuration to suppress grain boundary migration and crystalline reconstruction, which are the major mechanism underlying the instability of nanostructures[6,7,31,32]. The majority of multiple nanotwins were enclosed by low-surface-energy {111} and {100} facets, thus facilitating to lower the global free energy of Turing nanosheets[33]. Consequently, the facet-structure degradation and twin-free Wulff transformation are significantly suppressed by the Turing structure. In addition, the lamellar morphology of Turing PtNiNb had large surface contact and thus can be well anchored on the carbon supports, further preventing the detachment during the electrocatalytic process.

To further demonstrate the advantages of Turing structure in electrocatalysis, we tested the catalytic properties of the group of Pt-Ni-Nb nanosheets with different compositions discussed above (Pt content: 44.3–76.9 at%). Although these Pt-Ni-Nb nanosheets (without Turing structure) has significantly different platinum content, Turing PtNiNb shows the highest catalytic activity among all the prepared Pt-Ni-Nb catalysts (Supplementary Fig. 21). For example, the overpotentials of $Pt_{44.3}Ni_{42.1}Nb_{13.6}$ nanosheet and $Pt_{76.9}Ni_{17.8}Nb_{5.3}$ nanosheets are 47.8 mV and 29.1 mV at 10 mA cm$^{-2}$, larger than that of the Turing PtNiNb ($Pt_{56.1}Ni_{33.5}Nb_{10.4}$, 18.0 mV). Subsequently, the $Pt_{56.1}Ni_{33.5}Nb_{10.4}$ nanosheet without Turing structure was prepared as the reference sample (the same composition as Turing PtNiNb, the TEM images are shown in Supplementary Fig. 22a, b). LSV curves show that the catalytic activity of Turing PtNiNb is much higher than that of the PtNiNb nanosheet without Turing structure. The overpotentials of Turing PtNiNb and PtNiNb nanosheet without Turing structure are 18.0 mV and 127.5 mV at 10 mA cm$^{-2}$, respectively (Supplementary Fig. 22c, d). These comparisons suggest that Turing structure exhibits significant structural advantages in alkaline HER.

In order to evaluate the application potential of Turing catalyst for water electrolysis, we assembled an AEM water electrolyser by integrating Turing PtNiNb into a membrane-electrode-assembly (MEA) and using the PtIr gauze as the anode catalyst for oxygen evolution reaction (OER). The employment of PtIr gauze as the OER catalyst is to avoid the degradation in OER catalytic activity caused by the detachment of commercial Ir/C or $IrO_x$ powders during $O_2$ bubble evolution and electrolyte circulation, contributing to the evaluation of intrinsic stability of HER catalysts. The water splitting performance in Fig. 4g shows that the Turing PtNiNb|AEM|PtIr electrolyser exhibits a significantly improved catalytic activity compared with the Pt/C | AEM|PtIr electrolyser. The Turing PtNiNb-based electrolyser with a low Pt mass loading of 0.05 mg cm$^{-2}$ delivered much larger current densities than the Pt/C-based electrolyser (Pt mass loading ~ 0.3 mg cm$^{-2}$) at high applied voltages. Meanwhile, the cell voltage of the Turing PtNiNb-based electrolyser is only 1.98 V to reach the 1000 mA cm$^{-2}$ current density. The mass activity calculated from the Turing PtNiNb-based electrolyser is 20.758 A mg$^{-1}_{Pt}$ at 2.0 V, which is 23.5 times that of the Pt/C-based electrolyser (0.884 A mg$^{-1}_{Pt}$). Subsequently, the long-term stability of the Turing PtNiNb-based AEM water electrolyser was tested at the current density of 1000 mA cm$^{-2}$ (Fig. 4h). The Turing PtNiNb-based electrolyser maintained a near-constant voltage during the long-term operation. These results demonstrate that the Turing PtNiNb-based AEM water electrolyser can be stably operated at the current density of 1000 mA cm$^{-2}$ for at least 500 h without obvious degradation of catalytic activity.

## Theoretical calculation on Pt-Ni-Nb Turing configurations

The above analysis reveals Turing PtNiNb with the major crystallographic features of multiple nanotwins and lattice strain. We performed theoretical calculations to disclose the mechanism of enhanced HER activity induced by Turing structuring. Experimental data shows that the multi-fold nanotwins in Turing PtNiNb are mainly enclosed by {111} and {200} atomic planes with [110] zone axis. We cleaved the fcc $Pt_3Ni$ (110) layers and substituted Pt/Ni atoms to construct the four-atomic-layer fcc PtNi, fcc PtNiNb and fcc PtNiNb twin slab models. The fcc PtNi slab has a Pt: Ni ratio of 62.5: 37.5, which is almost consistent with the composition of bimetal $Pt_{62.6}Ni_{37.4}$ sample. The atomic proportions of the fcc PtNiNb and fcc PtNiNb twin slab models are the same: $Pt_{55.6}Ni_{33.3}Nb_{11.1}$, close to the experimental composition of Turing PtNiNb ($Pt_{56.1}Ni_{33.5}Nb_{10.4}$). Extensive structure optimization calculations were carried out to determine the stable slab configurations, and the results of fcc PtNiNb twin slab model are shown in Fig. 5a.

The hydrogen adsorption Gibbs free energy ($\Delta G_{H^*}$), a widely used descriptor for HER activity, was calculated to study the interaction between H$^*$ and catalysts[34]. The Nb doping and the incorporation of twin-configuration in fcc PtNiNb slab model greatly change the local atomic configurations (Supplementary Fig. 23 and 24). Figure 5b shows the summary of calculated values of $\Delta G_{H^*}$. The bimetal PtNi slab presents an optimal $\Delta G_{H^*}$ (0.046 eV, the bridge site of Ni-Ni) which is weaker than the $\Delta G_{H^*}$ of Pt (111) slab (−0.266 eV), revealing the positive contribution of Ni doping to catalytic activity. The Nb doping could greatly enrich the coordination types and optimize the $\Delta G_{H^*}$ of PtNiNb slab to -0.010 eV (the hollow site of Pt-2 shown in Supplementary Fig. 23). Moreover, five other active sites of the PtNiNb slab also shows the $\Delta G_{H^*}$ values between ± 0.200 eV (the top site of Nb, the bridge sites of Pt-Nb and Ni-Ni, the hollow sites of Pt-1 and Ni, as shown in Supplementary Fig. 23), suggesting the beneficial effects of Nb in improving the H$^*$ adsorption/desorption on catalyst surface. As for the PtNiNb twin slab, calculation results show the optimized $\Delta G_{H^*}$ of 0.047 eV which is larger than the smallest $\Delta G_{H^*}$ of the PtNiNb slab. Despite the inferior values of $\Delta G_{H^*}$, the incorporation of twin configuration in PtNiNb twin slab has yielded more active sites compared with the PtNiNb slab. For example, nine active sites of PtNiNb twin slab show the $\Delta G_{H^*}$ values between ± 0.200 eV and four active sites have the $\Delta G_{H^*}$ values between ± 0.100 eV. Specifically, the atoms on and near the twin boundary (the top site of Ni on TB; the top site of Nb near TB; the hollow site of Pt-2 on TB; as shown in Supplementary Fig. 24) contribute to the three of the most active sites. Twin configuration could result in partial symmetry breaking of a perfect fcc structure, and changes in orientation as well as interval of the atoms on sides of twin boundary. These crystalline features of twins are expected to induce significant changes in the local atomic coordination. The greatly increased number of active sites illustrate that the variation in atomic coordination induced by twin boundary can enormously improve catalytic activity. Hereafter, lattice strain was introduced in fcc PtNiNb slab. The statistical results on $\Delta G_{H^*}$ under compressive strain and tensile strain are shown in Fig. 5b. It is found that the compressive strain leads to the interaction optimization between H intermediates and catalyst surface. The PtNiNb twin exhibits the optimized $\Delta G_{H^*}$ values of −0.012 eV and about 0.000 eV under 1% and 2% compressive strain, respectively. This $\Delta G_{H^*}$ value (0.000 eV) of PtNiNb twin with 2% compressive strain (a hollow site of Pt on twin boundary) is close to the optimum value of $\Delta G_{H^*}$ for HER (0 eV), indicating the ideal moderate interaction between intermediates and active sites[34]. According to the Sabatier principle, this interaction, neither strong nor weak, is ideal for the adsorption/desorption of reactive participants[35]. This ideal interaction assures the desorption of H$^*$ to form $H_2$ facilely in the Tafel step. The weaker adsorption of hydrogen on Turing PtNiNb was further confirmed by CV measurements with the left shift of underpotentially deposited

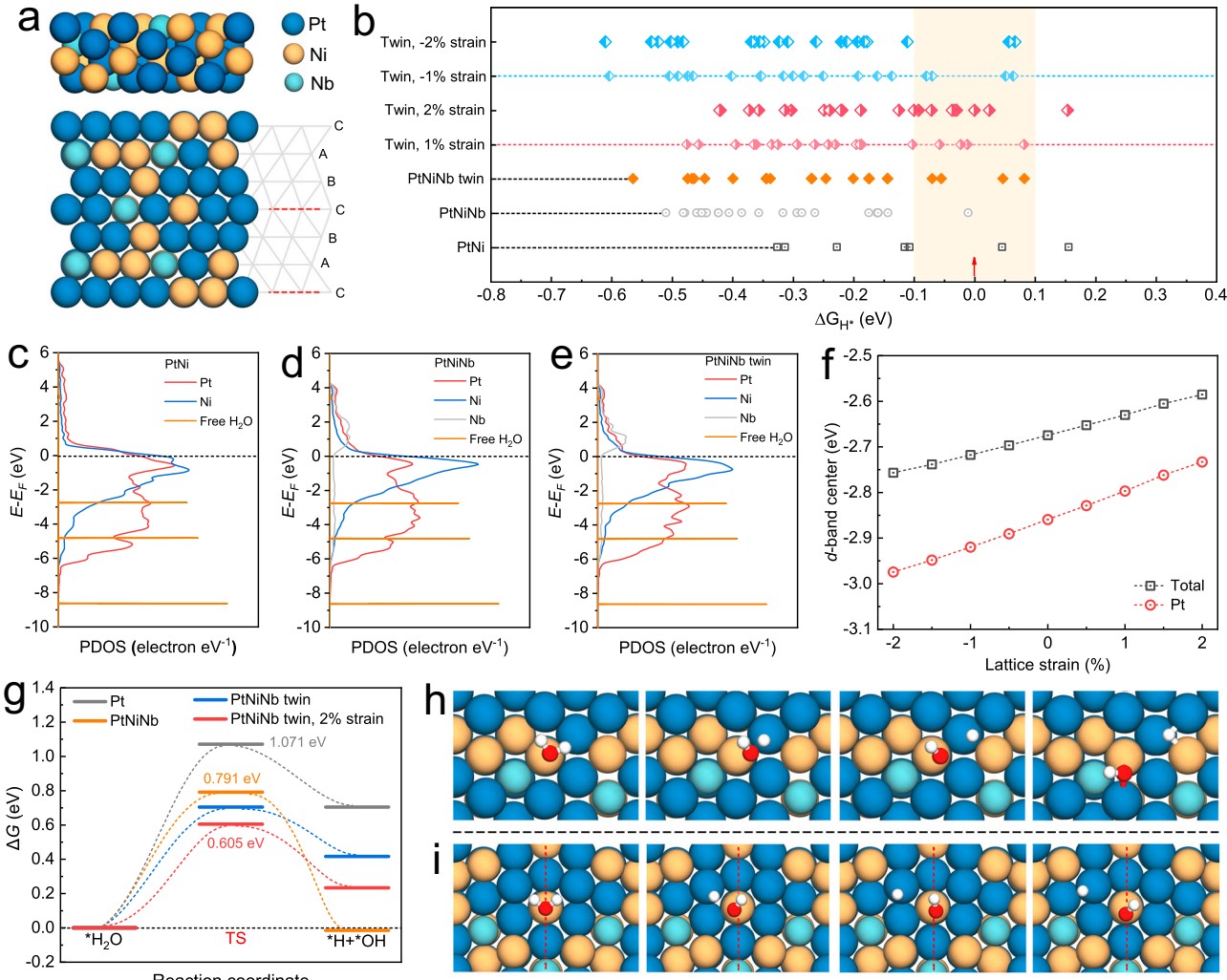

**Fig. 5 | Theoretical calculations. a** Top view and side view of PtNiNb calculation model, where the gray grid represents twins configuration and red dashed lines are twin boundaries. **b** The hydrogen adsorption Gibbs free energy ($\Delta G_{H^*}$) of bimetal PtNi, PtNiNb and PtNiNb twin models, the synergic effects of twin configuration and lattice strain on $\Delta G_{H^*}$ are investigated. The projected electronic density of states for $d$ orbital of PtNi **c**, PtNiNb **d** and PtNiNb twin **e**, including the PDOS diagram of free $H_2O$ molecule. **f** The $d$-band centers of PtNiNb twins as a function of tensile strains (positive) and compressive strains (negative). **g** Free-energy diagram for water dissociation on Pt, PtNiNb, PtNiNb twins and PtNiNb twins with 2% compressive strain. TS, transition state. The dynamic atom-configurations for water dissociation on the facet of PtNiNb **h** and PtNiNb twins with 2% strain **i**, including the $H_2O$ adsorption, the transition state and the final state.

hydrogen ($H_{upd}$) peak from 0.236 V vs. reversible hydrogen electrode (RHE) for Pt/C to 0.198 V vs. RHE for Turing PtNiNb (Supplementary Fig. 25)[36]. Furthermore, the PtNiNb twin with 2% compressive strain shows a significantly increased density of active sites: eight active sites with $\Delta G_{H^*}$ values between ±0.200 eV, which is 2-fold the PtNiNb twin without lattice strain. In this case, the atoms on the twin boundary contribute seven considerably active sites with the $\Delta G_{H^*}$ values between ± 0.100 eV (the top sites of Pt and Nb on TB; the hollow sites of Pt-1, Pt-2 and Pt-3 on TB; the hollow site of Pt-4 near TB; the bridge site of Ni-Ni-1 near TB; as shown in Supplementary Fig. 24). These results illustrate that twin boundary could provide extra active sites. In contrast, tensile strain results in the general reduction in $\Delta G_{H^*}$ of PtNiNb twin. The reduction in $\Delta G_{H^*}$ suggests the enhanced adsorption of reaction intermediates on the catalyst surface, thus slowing the $H^*$ conversion and $H_2$ evolution.

The optimized interaction between reaction intermediates and catalyst surface is attributed to the changes in electronic structure. The projected density of states (PDOS) of $d$ orbital were further analyzed and the results are shown in Fig. 5. Ni has a higher density near the Fermi level compared with Pt, and this trait indicates that Ni could

donate more free electrons to near atoms during HER. Nb shows a broad band, but relative low density of occupation states. Nb doping in PtNi results in downshift of $d$-band center from −2.82 eV to −2.94 eV for Pt sites (Fig. 5c, d). The downshift of $d$-band center indicates that the transferred electron would fill $d$-band, i.e., the antibonding electronic states. Therefore, Nb-doping could suppress the over-binding of $H^*$ on catalyst surface and weaken the $H^*$ adsorption, improving $\Delta G_{H^*}$ to optimal value. This result is in accordance with the optimized $\Delta G_{H^*}$ in comparison with PtNiNb (-0.010 eV) and PtNi (0.046 eV). The effects of twin configuration on electronic structure were also investigated. And the results show that the introduction of twin configuration in PtNiNb lead to an upshift of $d$-band center from -2.94 eV (PtNiNb without twin) to −2.86 eV (Fig. 5e). Such changes in $d$-band center provide the theoretical interpretation to the $\Delta G_{H^*}$ deterioration from -0.010 eV (PtNiNb without twin) to 0.047 eV (PtNiNb twin). Nevertheless, the $d$-band center of the atoms on twin boundary was downshifted by 0.13 eV due to the introduction of twin configuration. The twin boundary may thus provide more active sites for the catalysis. Furthermore, the shift of $d$-band center can be tailored by lattice strain. As shown in Fig. 5f, the $d$-band centers of PtNiNb twin and Pt sites are

monotonously shifted towards negative energy direction by increasing the compressive strain. These results are consistent with the findings of $\Delta G_{H^*}$ calculations.

In alkaline HER, proton generally originates from the dissociation of water and this reaction is always considered as the rate-determining step. The experimental results suggest that the HER on Turing PtNiNb surface follows the Volmer-Tafel pathway. The transition states (TS) were searched along the corresponding reaction pathway and the reaction barriers are calculated. We first calculated the possible adsorption configuration of $H_2O$ and OH. It is found that $H_2O$ is preferentially adsorbed on Ni and Nb sites, but OH could be more stably adsorbed on Ni sites. The initial state and final state are therefore designed with $H_2O$-adsorbed on Ni sites, and OH-adsorbed on Ni sites with H-adsorbed on Pt sites, respectively. Both the PtNiNb and PtNiNb twin are calculated using the similar reaction pathway (Fig. 5h, i). The water dissociation on the PtNiNb twin proceeded on the twin boundary due to the stable adsorption configuration. Calculation results demonstrated that the kinetic barrier of PtNiNb twin slab for $H_2O$ dissociation (0.7042 eV) was lower than those of PtNiNi slab (0.791 eV) and Pt (111) slab (Fig. 5g). Besides, lattice strain facilitated water dissociation and further reduced the kinetic barrier to 0.605 eV. The lower dissociation barrier of water could markedly assist the proton production from water and facilitate the subsequent $H_2$ evolution. Therefore, twin configuration and lattice strain can significantly contribute to the high catalytic activity of PtNiNb in alkaline HER.

Except for HER, activity improvement by Turing structuring should be observed in other electrocatalytic reactions due to the intrinsic optimization in microstructure via twinning and lattice strain, such as ethanol oxidation reaction. As shown in Supplementary Fig. 26a, b, Turing PtNiNb showed superior EOR activity in both acidic and alkaline solutions, the corresponding mass activity is 1.7 times and 2.0 times higher than those of commercial Pt/C catalyst. To confirm the generic significance of the structure design for different electrocatalysts, we further demonstrate the achievement of similar Turing structures in different metallic catalysts, including Ir–Ni–Nb, Pd–Ni–Nb, and Ag–Ni–Nb nanosheets (Supplementary Fig. 27a–c). These Turing catalysts also exhibited strikingly enhanced HER performance in terms of lower overpotentials, compared with commercial 20 wt% Ir/C, Pd/C, and Ag/C, respectively (Supplementary Fig. 27d–i).

## Discussion

Thus far, Turing patterns have mainly been observed in soft organic matter, and in only a few inorganic substances. This work demonstrated that Turing structure can be generated in low-dimensional solids at the nanoscale and be correlated to crystallographic defect-engineering and strain effects. The resulting Turing 2D nanosheets were highly electrocatalytically active and stable owing to the combination of high-density nanotwins and large lattice strain. This should lead to its use as a model for the development of other electrocatalytic materials for practical applications in energy sustainability. Thus, Turing structure represents a new paradigm for the design of high-performance low-dimensional nanocatalysts, demonstrating how the integration of defect modulation and strain effects can optimize such materials' stability and catalytic activity.

## Methods

### Materials and chemicals

Potassium hydroxide (KOH), copper sulfate pentahydrate ($CuSO_4·5H_2O$), and Nafion solution (5 wt%) were purchased from Sigma–Aldrich. Sulfuric acid ($H_2SO_4$), ethanol, and isopropanol were purchased from Anaqua. Pt/C (20 wt%), Ir/C (20 wt%), Pd/C (20 wt%), and Ag/C (20 wt%) catalysts were purchased from Premetek. Pure Argon gas (Purity: 99.995%) and pure hydrogen gas (Purity: 99.999%) were purchased from Linde HKO Ltd (Hong Kong). Carbon cloth and carbon fiber paper were purchased from CeTech and Toray industries,

respectively. PtIr gauze was purchased from Alfa-Aesar. All chemical reagents were utilized as received without further purification. Deionized water was purified through a Millipore system.

### Materials preparation

The precursor of Turing nanocatalyst was fabricated by magnetron sputtering. A layer of Si (99.999% purity) was first deposited on a glass substrate as the sacrificial layer. Afterward, the superthin metal films were deposited by co-sputtering the Pt target (99.95% purity) and NiNb target (99.9% purity). The deposition rate was 4 nm per minute and the substrate temperature was 300 K. In the sputtering process, argon pressure was $3 \times 10^{-3}$ Torr and substrate bias voltage was −80 V. For the synthesis of Turing PtNiNb nanocatalyst, the as-deposited film was firstly immersed in 2.0 M KOH solution for 60 min. The Si layer would be etched away and the free standing Turing film was exfoliated from the substrate. The resulting Turing films were collected by centrifugation and washed four times with the ethanol/water mixture and absolute ethanol to remove residual potassium hydroxide. Subsequently, the product was ultrasonicated at 273 K for 60 min to yield homogenous Turing PtNiNb nanosheets. The final Turing PtNiNb nanocatalysts were dispersed in absolute ethanol for further tests. The final Turing PtNiNb nanocatalysts were dispersed in absolute ethanol for tests on carbon cloth. The nanocatalysts in ethanol were mixed with carbon black by a 1 h sonication process to obtain 40 wt% catalyst ink. After drying to the powder, they were dispersed in a mixture of isopropanol and deionized water (v: v = 1:1) for the electrochemical testing on RDE.

### Characterization

The height profile of the film step was measured by atomic force microscopy (AFM, Dimension Icon, Bruker) under a tapping mode. The super-sharp AFM tip (SHR150, Budgetsensors) was employed to test the film thickness. Microstructural analysis was conducted on a JEM 2100 F FEG transmission electron microscopy operated at 200 kV acceleration voltage. High-resolution TEM and HAADF-STEM was conducted using the double aberration-corrected JEM-ARM300F2 at an acceleration voltage of 300 kV. The compositions of the catalysts were analyzed using ICP-OES (Optima 8000, PerkinElmer, USA) and inductively coupled plasma–mass spectroscopy (7700x, Agilent, USA). X-Ray diffraction (XRD) patterns of the samples were measured by X-ray diffractometer (Smartlab, Rigaku, Japan) with Cu Kα radiation. XPS with monochromatized Al Kα X-rays radiation (PHI Model 5802, PHI, USA) was used to investigate the surface electronic properties. The bonding energy (BE) calibration of the spectra was referred to the C 1 $s$ peak located at BE = 284.8 eV for the analysis.

The absorption spectra were recorded at the BL11B beamline of Shanghai Synchrotron Radiation Facility (SSRF). The XAFS data were collected in transmission mode using a Lytle ionization chamber (EXAFS Co.) filled with Ar. The beam current of the storage ring was 200 mA in a top-up mode. The incident photons were monochromatized by a Si(111) double-crystal monochromator, with an energy resolution $\Delta E/E \sim 1.4 \times 10^{-4}$. The spot size at the sample had dimensions of ~200 μm × 250 μm. The energy of the absorption edge ($E_O$) was calibrated using metal foil. The XAFS data were processed using Athena (version 0.9.26) and Artemis (version 0.9.26) software.

### Electrochemical measurements

The electrochemical measurements were performed at a CHI 660E electrochemical station (CH Instruments, USA) with a three-electrode electrochemical cell, using a carbon rod as the counter electrode and a reference electrode of Hg/HgO in alkaline electrolyte or saturated calomel electrode SCE in acidic electrolyte. Prepared catalyst inks were dropped on the carbon clothes (0.65 cm²) or RDE (0.196 cm²) as the working electrode connecting to the electrochemical workstation (The

catalyst mass loading is about 0.08 mg cm$^{-2}$ and the Pt mass loading is about 0.06 mg cm$^{-2}$). Ar-saturated KOH or H$_2$SO$_4$ solution was used as the electrolytes. All measurements were carried out at room temperature.

All LSV curves were recorded in 1.0 M KOH electrolyte at the scan rate of 5 mV s$^{-1}$ with 85% ohmic drop compensation. The resistances were measured by iR Comp module in CHI 660e electrochemical station at least 3 times before tests. C$_{dl}$ was measured by CV at scan rates of 10–80 mV s$^{-1}$ at non-faradaic overpotentials. The EIS measurements were carried out at an overpotential of 50 mV with a frequency range of 0.1 to 10$^5$ Hz and an amplitude of 5 mV. The accelerated stability test was performed by measuring the LSV curves before and after 30,000 continuous CV cycles in the potential range from −0.2 to 0.0 V vs. RHE with a sweep rate of 200 mV s$^{-1}$. Meanwhile, LSV curves after 10,000 and 20,000 cycles were collected for comparison. The durability of the catalyst was evaluated by the chronopotentiometry and the galvanostatic method.

Cu underpotential deposition (upd) measurements[37]. After electrochemical cleaning, the electrode-loaded catalysts were polarized at 0.255 V (vs. RHE) for 700 s in the 0.1 M H$_2$SO$_4$ containing 5 mM CuSO$_4$ electrolyte. Then the Cu stripping cycles were collected in the 0.1 M H$_2$SO$_4$ at a scan rate of 10 mV s$^{-1}$. Electrochemical active surface area (ECSA) of Turing PtNiNb, PtNi 2Dmetal and Pt 2D metal were calculated based on the Cu upd stripping CV curves and equation:

$$ECSA = \frac{Q_{Cu}}{0.42 \times [Pt]} = \frac{S_{Area}}{V_{scan} \times 0.42 \times [Pt]} \tag{1}$$

where $Q_{Cu}$ (mC) is the average charge integrated from Cu upd stripping process (0.3–0.8 V vs. RHE, Cu$_{upd}$ → Cu$^{2+}$ + 2e$^-$) on the CV curves, which can be determined by the ratio of integrating Cu upd stripping peaks area with the subtraction of the double layer to scanning velocity (10 mV s$^{-1}$). 0.42 mC cm$^{-2}$ is the electrical charge associated with monolayer adsorption of copper atoms on Pt, and [Pt] is the loading of Pt on the working electrode (0.0156 mg in our experiments).

The turnover frequency (TOF (s$^{-1}$)) can be calculated with the following equation:

$$TOF = \frac{I}{2Fn} \tag{2}$$

where $I$ is the current (A) during linear sweep measurement, $F$ is the Faraday constant (96485 C mol$^{-1}$), $n$ is the number of active sites (mol, $n = Q_{Cu}/2F$). The factor 1/2 is based on the consideration that two electrons are required to form one Cu atom.

Mass-normalized charge for H$_2$ evolution in stability measurement was calculated based on curves of galvanostatic and potentiostatic test and equation:

$$Mass - normalized\ charge = \frac{Q}{m_{PGM}} \tag{3}$$

Where $Q$ is the charge (C) which is obtained by integrating the $I$-$t$ curve (curves of decreasing current density simplify the integration to calculate the trapezoidal area), $m_{PGM}$ is the mass loading of platinum-group-metal in the stability measurement.

## MEA fabrication and measurement
Integrating Turing PtNiNb catalyst into an MEA was used to construct an AEM water electrolyser device and evaluate its electrochemical performance using Autolab PGSTAT 302 N. An anion exchange membrane Fumasep FAA-PK-130 (Fumasep, German) was immersed in 1.0 M KOH solution for 24 h to exchange Cl$^-$ into OH$^-$ before being used in MEA. The catalyst ink was prepared by

homogeneously dispensing 1 mg of Turing PtNiNb catalyst into an ethanol and FAA-PK-130 solution mixture under sonication at 0 °C for about 1 h. Afterward, 0.16 ml of catalyst ink was sprayed on the carbon fiber paper with an exposed surface area of 1 cm$^2$. (The mass loading is about 0.05 mg$_{Pt}$ cm$^{-2}$) The carbon fiber paper was fixed on a hotplate whose temperature was 95 °C, ensuring that the solvent evaporated quickly. A commercial Pt/C cathodic electrode was also prepared in the above method using Pt/C (20 wt% Pt, Premetek, USA) and the mass loading is about 0.3 mg$_{Pt}$ cm$^{-2}$. The PtIr gauze (Alfa Aesar, USA) was the anodic catalyst layer. The cathodic part, anodic part and as-prepared anion exchange membrane were integrated into an AEM water electrolyser device and evaluated its water electrolysis performance. Flow fields were also sealed with electrically insulating gaskets to prevent liquid and gas from escaping. By using a peristaltic pump, 1.0 M KOH was circulated through the cathodic and anodic sides at room temperature. The polarization curves were measured at 25 °C under ambient pressure from 1.2 V to 2.2 V. A stability test was performed using galvanostatic electrolysis at 1.0 A constant current at room temperature. The galvanostatic electrolysis was held for 30 min to achieve a steady state before obtaining the stability curves.

## DFT models and calculations
Different models were constructed based on bulk-relaxed cells of Pt, PtNi, and PtNiNb. The Pt (111) surface was modeled by four-layer periodic slabs with (3 × 3) supercells. For the model of PtNiNb twins, 4-layered (110) slabs of PtNiNb were created with 72 atoms in total and two parallel Σ3 {111} TBs in a single cell. The atoms in the top two layers were fully relaxed in calculations, while the atoms in the bottom two layers were fixed for each model. A vacuum slab of 15 Å was added along the z-axis in each model to avoid periodic interactions. The Brillouin zone was sampled with the gamma-centered Monkhorst-Pack scheme of 3 × 3 × 1, 3 × 3 × 1, and 2 × 2 × 1 k-point mesh for models of Pt (111), PtNi and PtNiNb twins (PtNiNb), respectively. DFT calculations were performed by using Vienna Ab-initio Simulation Package with the projector-augmented wave pseudopotentials[38–41]. The electron exchange-correlation energy was described by the Perdew−Burke−Ernzerhof functional within the generalized gradient approximation[42]. The energy cutoff of the plane-wave basis set was 400 eV. The force convergence criterion for atom relaxation was 0.05 eV Å$^{-1}$. The climbing image nudged elastic band method was employed to search the transition state and determine the reaction barrier for water dissociation on these catalyst surfaces. The Gibbs free energy change for atomic hydrogen adsorption ($\Delta G_{H^*}$) was calculated as:

$$\Delta G_{H^*} = \Delta E_{H^*} + \Delta ZPE - T\Delta S_H \tag{4}$$

Where $\Delta G_{H^*}$ is the hydrogen adsorption energy; $\Delta ZPE$ and $\Delta S_H$ are the zero-point energy and entropy difference between the adsorbed state and gas state; $T$ is the temperature. $\Delta E_{H^*}$ was calculated as:

$$\Delta E_{H^*} = E_{(slab+H)} - E_{slab} - \frac{1}{2}E_{H_2} \tag{5}$$

Where $E_{(slab+H)}$ is the energy of the model with a hydrogen atom adsorbed on surface; $E_{slab}$ is the energy of the corresponding model without any adsorbate; $E_{H_2}$ is the energy of a hydrogen molecule in gas state.

## Data availability
The data generated in this work are presented in this article and the Supplementary Information, and are available from the corresponding authors upon request.

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

## Acknowledgements

This work was supported by the National Key R&D Program of China (Project No. 2017YFA0204403); the Guangdong Provincial Department of Science and Technology (Key-Area Research and Development Program of Guangdong Province) under grant 2020B090923002; Shenzhen-Hong Kong Science and Technology Innovation Cooperation Zone Shenzhen Park Project: HZQB-KCZYB-2020030; Shenzhen Science and Technology Program (JCYJ20220818101204010); GRF- RGC General Research Fund Scheme (Ref. CityU 11216219); The Innovation & Technology Commission of HKSAR, through the Hong Kong Branch of National Precious Metals Material Engineering Research Center; and the Ministry of Science and Technology of China (Project No.

2018YFE0190400). This work made use of the resources of the TRACE TEM center at City University of Hong Kong. The authors thank BL11B beamline of the Shanghai Synchrotron Radiation Facility (SSRF) for providing the beamtime.

## Author contributions

J.L. supervised the project. J.L.G. conceived the idea and concept. J.L.G. and L.X.L. designed the experiments and interpretation of the results. J.L.G. prepared the catalysts and performed structure characterizations, surface morphology tests, and theoretical interpretation. L.X.L. conducted the electrocatalytic, X-ray diffraction, and composition analyses. J.L.G. and F.B.T. performed DFT simulations. B.C. and Y.N.X. contributed to the TEM characterizations. J.Z., J.D.S., and Y.J.W. contributed to the data analysis. J.L.G. and L.X.L. wrote the manuscript. All authors participated in the discussion of the results.

## Competing interests

The authors declare no competing interests.
