## [Peer Review File · Nature Communications]

REVIEWER COMMENTS

Reviewer #1 (Remarks to the Author):

In this work, Lu and coworkers reported a Turing structure of PtNi superthin nanosheets synthesized by introducing high-density nanotwins. It indicated that Turing structure can be formed at the nanoscale in low-dimensional solids and be correlated to crystallographic defect-engineering and strain effects. The as-synthesized Turing structures are well characterized and analyzed. The resulting Turing 2D nanosheets were highly electrocatalytically active and stable owing to the combination of high-density nanotwins and large lattice strain. This strategy could be developed for synthesizing other electrocatalytic materials for industrial applications. In my opinion, this paper could be published in Nature Communications, but major revision should be further made.

(1) The authors only synthesized a Turing PtNi with Pt_{56.1}Ni_{33.5}Nb_{10.4}. However, I think that the composition of alloy should have a great effect on the Turing structure. The authors should provide related experiments to approve this point.

(2) I notice that Nb as the major elements is used for the synthesis of Turing structure. However, when the authors discussed the catalytic activity of Turing, they neglected the effect of Nb. Why? Metal doping should affect the activity. Thus, the authors provided more discussion and experiments on this.

(3) Ni and Nb alloying Pt has a significant effect on the chemical environment of Pt active sites. Lots of researches have been reported that metal doping could improve the HER activity of Pt. The authors have to analyze it in detail using the XAFS, and analyzed their relationship to the HER activity.

(4) The Turing structures have lots of edge structures with different defects. Generally, these defects also could improve the electrocatalytic activity. The authors should enhance the discussion on it.

Reviewer #2 (Remarks to the Author):

The manuscript by Lu et al. reported the superthin PtNiNb nanosheets with unique Turing stripes. The morphology regulation of Pt-based nanomaterials is crucial for the enhancement of atomic utilization rate, active site density and exposed specific active planes. In this work, the as-described Turing structure is quite interesting, whose spatiotemporal stationary pattern and concomitant high-density nanotwins/large lattice strain endow it with high intrinsic activity and stability. However, three significant problems were important for the integrity and persuasiveness of this work, but not well explained which let me reject it to be published in Nature Communications :

1. Why could the ordinary two-step magnetron sputtering followed by the chemical etching process result in the distinctive Turing structure? Is it affected by the Si sacrificing temple? These issues need more supplemental evidences, contrast experiments and theoretical bases to expound the principle, feasibility and universality of the as-stated strategy.
2. The valence bond structure and atomic coordination information are missing.
3. The theoretical modes used in this work is inconsistent with the actual structure. The main product is Pt_{56.1}Ni_{33.5}Nb_{10.4} with the material phase of fcc Pt/PtNi. But the DFT model uses a fcc Pt₃Ni. The effect of Nb element is disregarded and the elemental composition is totally different. That is definitively wrong. With the reasonable model, the crystal structure of pristine crystal plane, and the planes with incorporated twins and strain should be exhibited, in order to explain how the twins and strain are introduced. Furthermore, 1) what kind of strain is exerted? Tensile or compressive strain? 2) Why did the Pt 2D metal use the Pt(111) as calculation model? 3) The electronic structure calculations need to be supplemented.

The above three problems make the structure-activity unconvincing, thus, I cannot support its publication in Nature Communications.

Some small issues are listed as follows:

- a) Why was the "PtNiNb nanosheets" abbreviated to "Turing PtNi"? The Nb is totally ignored. And the actual bimetal PtNi contrast sample is missing, which is very important in this work, especially for the electrochemical evaluation.
- b) Was the synthesis method for the contrastive Pt and NiNb (as well as the missing PtNi) the same as the main sample? Did the Turing structure also exist in these samples?
- c) Electrochemical impedance spectroscopy should be supplemented to offer the solution resistance and charge transfer resistance.
- d) Could the chemical etching process (immersed in 2 M KOH for 60 min) totally remove the Si layer?
- e) The catalyst ink is more suitable for the glassy carbon electrode measurement. Within the low current density range in this work, the glassy carbon electrode is not restricted by evident mass transfer problem, and can better reflect the intrinsic activity. Even with the carbon cloth electrode, the actual reaction area and the loading were not provided as well.

Reviewer #3 (Remarks to the Author):

The authors report Turing structured PtNiNb electrocatalysts for hydrogen evolution reaction in an alkaline medium. The work looks like the authors have endeavoured to organize this manuscript from the good level of completion of this article (e.g., in-depth analysis for physical characterizations,

performance evaluation, DFT, and post-testing studies). Especially the structural/topological studies on PtNiNb materials provide an in-depth understanding of the materials for lattice distortion, defects, twinning, or 3D architectures. I can agree that the material preparation & corresponding physical analyses are comprehensive in this study but the major claims and the structural novelty are less studied or not well-supported by experimental evidence or computational approaches. Some major comments that should be addressed are listed as follows to consider this article publishing in Nature Communications:

1. The catalyst named PtNi is Pt_{56.1}Ni_{33.5}Nb_{10.4} from ICP-OES analysis. What is the role of Ni & Nb atoms for this structural aspect and the electrochemical role?
2. The authors have claimed many aspects regarding structural analyses. For example, 5-fold & 2-fold twins, stacking faults and lattice distortion on the PtNiNb sample. However, the modelling of DFT suggests Pt-Ni sample without Nb inside. What is the reason why the actual materials and the DFT model presents different structure? Please provide the DFT results for the actual model.
3. If the sample can be prepared by a thin-film type (authors have claimed), why the sample had to be dropped-coated on the carbon paper support? Because the nanostructure of the Turing PtNiNb is the major novelty of the work, I guess the sample film had to be studied by transferring the other (plate-like) substrate. Please provide if the drop-casted sample can present identical structural benefits compared to a thin-film type of sample.
4. The authors claimed that the strain effect comes from the twins or defects can be beneficial for this Turing system. Then, where are the active sites for the reaction? The 5-fold twins or defective sites themselves cannot be active sites? What is the role of Ni and Nb in the electrochemical aspect?
5. Is the sample controllable for the atomic composition? Since the authors claimed that the topological properties can be beneficial for the Turing catalyst system and provided DFT calculations to support them. More electrochemical/experimental evidence needs to be provided by different compositional PtNiNb Turing samples.
6. Please provide the acidic HER performance.

Reviewers' comments:

Reviewer #1 (Remarks to the Author): In this work, Lu and coworkers reported a Turing structure of PtNi superthin nanosheets synthesized by introducing high-density nanotwins. It indicated that Turing structure can be formed at the nanoscale in low-dimensional solids and be correlated to crystallographic defect-engineering and strain effects. The as-synthesized Turing structures are well characterized and analyzed. The resulting Turing 2D nanosheets were highly electrocatalytically active and stable owing to the combination of high-density nanotwins and large lattice strain. This strategy could be developed for synthesizing other electrocatalytic materials for industrial applications. In my opinion, this paper could be published in Nature Communications, but major revision should be further made.

(1) The authors only synthesized a Turing PtNi with Pt_{56.1}Ni_{33.5}Nb_{10.4}. However, I think that the composition of alloy should have a great effect on the Turing structure. The authors should provide related experiments to approve this point.

Answer: Thanks for the reviewer's comments. We share this view that the catalyst composition may determine the formation of Turing structure. We conducted additional experiments to clarify the composition-structure relations in Pt-Ni-Nb metallic catalysts. Platinum is a widely studied active element for HER. Platinum content is speculated to be a significant contributor to the catalytic activity. Thus, we first investigate the effects of platinum content on the structure of Pt-Ni-Nb catalyst. Meanwhile, the ratio of nickel to niobium is fixed at approximately 3:1. These new nanocatalysts and Turing PtNiNb nanocatalyst were synthesized using the same method. The catalyst compositions were analysed by inductively coupled plasma–optical emission spectroscopy (ICP-OES). The microstructures of these nanocatalysts are tested by an aberration-corrected transmission electron microscope and the results are shown in Response-Figure 1. Pt_{56.1}Ni_{33.5}Nb_{10.4} catalyst (denoted as Turing PtNiNb) presents the clear characteristics of Turing structure which is the same as previous findings (Response-Figure 1d). As the platinum content decreasing from 56.1 at.% to 51.7 at.%, the stripes tend to grow in-plane and parts of coarsening stripes interconnected to form large junctions (Response-Figure 1c). Furthermore, the traits of Turing structure eventually

disappeared in $\text{Pt}_{44.27}\text{Ni}_{42.11}\text{Nb}_{13.62}$ sample (Response-Figure 1a and b). However, higher platinum contents could also cause the breakdown of Turing structure. As shown in Response-Figure 1, $\text{Pt}_{60.1}\text{Ni}_{30.8}\text{Nb}_{9.1}$, $\text{Pt}_{69.2}\text{Ni}_{23.6}\text{Nb}_{7.2}$, $\text{Pt}_{73.4}\text{Ni}_{20.1}\text{Nb}_{6.5}$ and $\text{Pt}_{76.9}\text{Ni}_{17.8}\text{Nb}_{5.3}$ samples are planar nanosheets without the features of Turing structures. All the high-platinum-content samples (≥ 60.1 at.%) are composed of classical nanocrystals and do not have particular topographical characteristics. These results suggest that the composition of Turing structure seems to be limited to a narrow range.

Moreover, reviewers are greatly interested in the effects of niobium doping on structures. We therefore remove niobium element from the Pt-Ni-Nb alloy and synthesized binary $\text{Pt}_{63.7}\text{Ni}_{36.3}$ nanocatalyst. Binary $\text{Pt}_{63.7}\text{Ni}_{36.3}$ catalyst has a Pt-Ni ratio of 1.75 that is close to the ratio of Pt to Ni (1.67) in Turing PtNiNb ($\text{Pt}_{56.1}\text{Ni}_{33.5}\text{Nb}_{10.4}$). TEM images of the $\text{Pt}_{63.7}\text{Ni}_{36.3}$ catalyst show the nanocrystals with random orientations (Response-Figure 2). These nanosized crystals are densely arranged with clear grain boundaries. This kind of classical morphology of nanometallic materials is similar to the microstructures of the high-platinum-content samples discussed above and is sharply contrast to Turing structure. The niobium-deletion in Pt-Ni alloys would lead to the collapse of Turing structure and this experiment underlines the important role of niobium to induce the formation of Turing structure.

The revisions corresponding to this comment are added in Page 6, Paragraph 2; Page 7, Paragraph 1; Supplementary Fig. 4 and Supplementary Fig. 5 of the revised manuscript.

Response-Figure 1 a, The HRTEM image of Pt_{44.3}Ni_{42.1}Nb_{13.6} nanosheet. TEM images of Pt_{44.3}Ni_{42.1}Nb_{13.6} nanosheet (**b**), Pt_{51.7}Ni_{37.0}Nb_{11.3} nanosheet (**c**), Turing PtNiNb (**d**), Pt_{60.1}Ni_{30.8}Nb_{9.1} nanosheet (**e**), Pt_{69.2}Ni_{23.6}Nb_{7.2} nanosheet (**f**), Pt_{73.4}Ni_{20.1}Nb_{6.5} nanosheet (**g**), Pt_{76.9}Ni_{17.8}Nb_{5.3} nanosheet (**h**). **i**, The HRTEM image of Pt_{76.9}Ni_{17.8}Nb_{5.3} nanosheet.

Response-Figure 2 a. The TEM image of bimetal PtNi nanosheet. **d**. The HRTEM image of bimetal PtNi nanosheet.

(2) I notice that Nb as the major elements is used for the synthesis of Turing structure. However, when the authors discussed the catalytic activity of Turing, they neglected the effect of Nb. Why? Metal doping should affect the activity. Thus, the authors provided more discussion and experiments on this.

Answer: Thanks for the reviewer reminding us of the Nb-doping effects on catalytic activity. Element doping has been frequently demonstrated as an efficient strategy to improve catalytic activity. Niobium takes up 10.3 at.% in Turing PtNiNb sample and the effects of niobium doping on catalytic activity of the Turing catalyst should be analysed in detail. In the response to comments (1) of reviewer #1, we found that Nb-deletion resulted in the breakdown of Turing structure for Pt-Ni-Nb alloys. Such structure transition from Turing structure to classical nanocrystalline materials definitely cause changes in catalytic activity. In this section, we tested the catalytic performance of binary $\text{Pt}_{63.7}\text{Ni}_{36.3}$ catalyst and conducted the comparison between $\text{Pt}_{63.7}\text{Ni}_{36.3}$ and Turing PtNiNb ($\text{Pt}_{56.1}\text{Ni}_{33.5}\text{Nb}_{10.4}$). It should be noted that $\text{Pt}_{63.7}\text{Ni}_{36.3}$ and Turing PtNiNb have a close ratio of platinum to nickel. LSV curves show the higher catalytic activity of Turing PtNiNb in comparison to that of the binary $\text{Pt}_{63.7}\text{Ni}_{36.3}$ catalyst (Response-Figure 3a). The overpotentials at 10 mA cm^{-2} of Turing PtNiNb and $\text{Pt}_{63.7}\text{Ni}_{36.3}$ catalyst are 18.0 mV and 27.2 mV, respectively. Besides, the mass activity of Turing PtNiNb ($7166.3 \text{ mA cm}^{-2}$) is about 6.1-fold that of the $\text{Pt}_{63.7}\text{Ni}_{36.3}$ catalyst ($1179.2 \text{ mA cm}^{-2}$). Turing PtNiNb has a smaller Tafel slope (32.7 mV dec^{-1}) than $\text{Pt}_{63.7}\text{Ni}_{36.3}$ catalyst (35.3 mV dec^{-1}), suggesting the more efficient catalytic kinetics (Response-Figure 3b). Furthermore, we evaluated the catalytic stability of binary $\text{Pt}_{63.7}\text{Ni}_{36.3}$ catalyst using the chrono-potentiometric test and the accelerated durability test. Turing PtNiNb nanocatalyst is considerably stable in these harsh stability test. In contrast, $\text{Pt}_{63.7}\text{Ni}_{36.3}$ catalyst shows a large overpotential change of 170.5 mV and pronounced decline in current density (Response-Figure 3c and d). The above results show that niobium doping greatly improve the catalytic activity and durability of Pt-Ni catalysts.

We further analyse beneficial effects of Nb-doping on catalytic activity by DFT calculations. An fcc PtNiNb slab model ($\text{Pt}_{55.6}\text{Ni}_{33.3}\text{Nb}_{11.1}$) without twin configuration was constructed. The fcc PtNi slab model has a Pt : Ni ratio of 62.5 : 37.5, which is similar to the Pt:Ni ratio in the fcc PtNiNb slab model. The hydrogen adsorption Gibbs free energy (ΔG_{H^*}), a widely used descriptor for HER activity, was calculated to study the interaction between H^* and catalysts. Response-Figure 4a shows the summary of calculated values of ΔG_{H^*} . The bimetal PtNi slab presents an optimal ΔG_{H^*} (0.046 eV, the bridge site of Ni-Ni) which is weaker than the ΔG_{H^*} of Pt (111) slab (-0.266 eV). Moreover, the bimetal PtNi slab has four active sites with the ΔG_{H^*} values between ± 0.200 eV. The Nb doping could greatly enrich the coordination types and optimize the ΔG_{H^*} of PtNiNb slab to -0.010 eV (the hollow site of Pt-2 shown in Supplementary Fig. 23 of the revised manuscript). And five other active sites of the PtNiNb slab also show the ΔG_{H^*} values between ± 0.200 eV (the top site of Nb, the bridge sites of Pt-Nb and Ni-Ni, the hollow sites of Pt-1 and Ni, as shown in Supplementary Fig. 23 of the revised manuscript), suggesting the beneficial effects of Nb in improving the H^* adsorption/desorption on catalyst surface. The projected density of states (PDOS) of d orbital were further analysed and the results are shown in Response-Figure 4b and c. Nb doping in PtNi results in downshift of d -band centre from -2.82 eV to -2.94 eV for Pt sites. The downshift of d -band centre indicates that the transferred electron would fill d -band, i.e., the antibonding electronic states [B. Hammer et al., *Nature* 376 (1995), 238] [J. K. Norskov et al., *Proc. Natl. Acad. Sci. U. S. A.* 108 (2011), 937]. Therefore, Nb-doping could suppress the over-binding of H^* on catalyst surface and weaken the H^* adsorption, improving ΔG_{H^*} to optimal value. This result is in accordance with the optimized ΔG_{H^*} of PtNiNb (-0.010 eV) in comparison with PtNi (0.046 eV).

The revisions corresponding to this comment are added in Page 16, Paragraph 1; Page 25, Paragraph 2; Page 27, Paragraph 2; Fig. 5 of the revised manuscript.

Response-Figure 3 a, LSV curves of Turing PtNiNb and PtNi 2D metal in 1.0 M KOH. RHE, reversible hydrogen electrode. **b**, Tafel plots of Turing PtNiNb and PtNi 2D metal. **c**, Long-term stability test of Turing PtNi and PtNi 2D metal under a large current density of 200 mA cm^{-2} in 1.0 M KOH. **d**, Variation of current density during accelerated durability test. Columns refer to values at 60 mV recorded in LSV curves prior to testing, and after 10,000 (10k), 20,000 (20k) and 30,000 (30k) cycles.

Response-Figure 4 a, The hydrogen adsorption Gibbs free energy (ΔG_{H^*}) of bimetal PtNi and PtNiNb models. The projected electronic density of states for d orbital of PtNi (b) and Turing PtNiNb (c), including the PDOS diagram of free H₂O molecule.

(3) Ni and Nb alloying Pt has a significant effect on the chemical environment of Pt active sites. Lots of researches have been reported that metal doping could improve the HER activity of Pt. The authors have to analyze it in detail using the XAFS, and analyzed their relationship to the HER activity.

Answer: Thanks for the reviewer's comments. In the above response to comment (2), we provide experiment results to clearly verify the benefits of Nb doping on catalytic performance of Pt-Ni-Nb alloys. In this section, we experimentally analysed the effects of Ni doping and the X-ray absorption spectroscopy experiments. Several

studies have reported that bimetallic Pt-Ni alloys are highly active catalysts for HER. The high activities originated from the synergetic effects of Ni/NiO_x which accelerate water dissociation and the conversion of H* to H₂ [P.T. Wang et al., *Angew. Chem. Int. Edit.*, 55 (2016), 12859]. In this work, we also reveal the high catalytic activity of binary Pt_{63.7}Ni_{36.3} catalyst that has a similar ratio of Pt to Ni in Turing PtNiNb. As shown in Response-Figure 5a, the LSV curves demonstrate a smaller overpotential of binary Pt_{63.7}Ni_{36.3} catalyst at 10 mA cm⁻² (27.2 mV) compared with the Pt 2D metal (33.6 mV) and commercial 20 wt.% Pt/C (51.0 mV). The Pt_{63.7}Ni_{36.3} catalyst shows a lower Tafel slope than Pt 2D metal and commercial 20 wt.% Pt/C (Response-Figure 5b). Both the Pt_{63.7}Ni_{36.3} catalyst and Pt 2D metal are synthesized by the same method in the form of nanosheets. The above comparisons illustrate the beneficial effects of Ni doping on catalytic performance for Pt-based catalyst design.

Response-Figure 5 a, LSV curves of PtNi 2D metal, Pt 2D metal and Pt/C in 1.0 M KOH. RHE, reversible hydrogen electrode. **b**, Tafel plots of PtNi 2D metal, Pt 2D metal and Pt/C.

Furthermore, we have additionally conducted the X-ray absorption spectroscopy experiments, and carefully discussed the valence bond structure as well as atomic coordination information. **The following content and Figures were added in the revised manuscript to reveal XAS results:**

The Turing PtNiNb presents a class of novel topological structure in nanometallic materials. We performed X-ray absorption near-edge structure (XANES) and extended

X-ray absorption fine structure (EXAFS) experiments to reveal the electronic structure and coordination information of Turing structure. The Pt L₃-edge XANES spectra shows that the absorption thresholds and white line peaks of the tested samples Turing PtNiNb (Pt_{56.1}Ni_{33.5}Nb_{10.4}), Pt_{56.1}Ni_{33.5}Nb_{10.4} (PtNiNb, without Turing structure), bimetal Pt_{63.7}Ni_{36.3} and Pt 2D metal are close to those of the Pt foil, suggest the Pt valence states are almost identical to zero (Response-Figure 6a below). The Fourier-transformed EXAFS spectra of Pt foil and Pt 2D metal shows the similar coordination environment, and the corresponding dominating peaks are assigned to Pt-Pt (~2.77 Å) coordination (Response-Figure 6d below). Ni doping results in the left shift of the dominating peak in the EXAFS spectra of binary PtNi sample, which is resolved as the Pt-Ni (2.62 Å) and Pt-Pt (2.69 Å) coordination. In contrast, Turing PtNiNb and PtNiNb have greatly different EXAFS spectra with significantly reduced amplitude of the atomic shell peaks. The fitting results of EXAFS spectrum reveal the major coordination of Pt-Pt (2.72 Å), Pt-Ni (2.59 Å) and Pt-Nb (2.67 Å) in Turing PtNiNb sample. The changes in peaks amplitude suggest the reduced coordination number. The fitting results show that the coordination number values of both Turing PtNiNb (8.1) and PtNiNb (7.4) are lower than those of bimetal PtNi (9.5), Pt 2D metal (9.5) as well as Pt foil (12, Supplementary Table 1 of the revised manuscript). Such pronounced reduction in coordination number is attributed to the small nanosheet size/thickness, fertile edge structures and substantial crystalline defects, including the nanotwins and stacking faults. The Pt coordination environment was further analysed by wavelet transform (WT) spectra at Pt L₃-edge. The WT spectrum in Response-Figure 6g shows that the maximum peak intensity of the Pt foil is located at 10.05 Å⁻¹, which originates from the Pt-Pt bond. The maximum peak intensity of bimetal PtNi shifts to 7.35 Å⁻¹ due to the contribution of Pt-Ni bond (Response-Figure 6h). Furthermore, the Pt-M paths of Turing PtNiNb shifts to a lower k space, in sharp contrast to the Pt-Pt coordination in Pt foil (Response-Figure 6i).

Response-Figure 6b shows the normalized Ni K-edge XANES spectrum. A clear pre-edge around 8333 eV was detected, suggesting the dipole-forbidden, quadrupole-

allowed transition ($1s$ to $3d$) [N. Becknell et al., *J. Am. Chem. Soc.*, 137 (2015), 15817]. The absorption edge of the XANES spectrum was applied to evaluate the Ni valence. Turing PtNiNb presents a higher adsorption threshold and a greater intensity of white line compared with the Ni foil, indicating the variations in electronic structure. Ni in Turing PtNiNb is speculated to be positively charged and parts of nickel species are oxidation states. The Ni valence states were further confirmed with the Ni 2p high-resolution X-ray photoemission spectra (XPS) showing the characteristic peaks of Ni^{2+} (Supplementary Fig. 8 of the revised manuscript). The Fourier transformed EXAFS spectra of bimetal PtNi sample shows a dominating peak at around 2.2 Å, which originates from the first shell of Ni-Ni coordination (Response-Figure 6e). The coordination environments of bimetal PtNi highly resemble those of Ni foil and NiNb nanosheet. Taking the bimetal PtNi as the reference, Nb doping contribute to the left shift in the dominating peaks of PtNiNb. Fitting results on the EXAFS spectrum of PtNiNb illustrate that the major coordination is Ni-Ni and the second peak locating around 2.65 Å can be attributed to the Ni-Nb coordination. By contrast, Turing PtNiNb present two distinct scattering paths in the EXAFS spectrum. The first peak at ~ 1.6 Å is assigned to Ni-O coordination, while the second peak at ~ 2.8 Å is speculated to originate from the second coordination shell of the Ni-Ni scattering path. The Ni-Pt coordination with a small coordination number of 0.6 was also resolved from the fitting of EXAFS spectrum (Supplementary Table 2 of the revised manuscript).

We also analyzed the XANES and EXAFS spectra of these sample at Nb K-edge. Turing PtNiNb, PtNiNb (without Turing structure) and NiNb samples show higher near-edge absorption energies and larger intensity of white line peaks than those of Nb foil, suggesting the higher Nb valence states in these three samples (Response-Figure 6c). The Fourier transformed EXAFS spectra of Turing PtNiNb and NiNb depicts the peak components between 1.2 Å and 1.9 Å as a result of Nb-O scattering pair, while this peak is relatively weak in PtNiNb sample (Response-Figure 6f). Furthermore, fitting calculations on the EXAFS spectrum of Turing PtNiNb resolved the Nb-Nb, Nb-Pt, Nb-Ni as well as Nb-O coordination. Turing structuring induced a significant

reduction in coordination numbers. The coordination number of Nb in Turing PtNiNb (4.6) is even smaller than that of amorphous NiNb nanosheet (7.0) which represents a kind of low-coordination structure (Supplementary Table 3 of the revised manuscript).

The greatly reduced coordination number of Ni and Nb elements in Turing PtNiNb indicate that the edge structures of Turing configuration, including the surface of the irregular stripes, interfaces and twin boundaries, have abundant low-coordinated atoms (Supplementary Table 2 and 3 of the revised manuscript). These findings are also consistent with the features of stacking-fault-activated Ag nanoparticle reported previously, in which stacking fault leading to a low coordination number [Z. Li et al., *Nat. Catal.*, 2 (2019), 1107]. In addition, the reduced coordination number could be attributed to the significantly increased proportion of surface atoms on the Turing stripes (the coordination number of surface atoms are generally lower than that of the inner atoms). This conclusion can be further supported by the greatly increased electrochemical active surface area of Turing PtNiNb (Supplementary Table 6 of the revised manuscript).

The revisions corresponding to this comment are added in Page 12, Paragraph 2; Page 13 and Page14; Fig. 3 of the revised manuscript.

Response-Figure 6 Electronic structure analysis. XANES spectra collected at Pt L₃-edge (a), Ni K-edge (b) and Nb K-edge (c). Fourier transformed EXAFS spectra at Pt L₃-edge (d), Ni K-edge (e) and Nb K-edge (f). Wavelet transforms analysis of the Pt L₃-edge for Pt foil (g), PtNi 2D metal (h) and Turing PtNiNb (i).

(4) The Turing structures have lots of edge structures with different defects. Generally, these defects also could improve the electrocatalytic activity. The authors should enhance the discussion on it.

Answer: Thanks for the reviewer reminding us to explore the effects of edge structure on electrocatalytic activity. Metallic catalysts are usually prepared with substantial edges structures, including the corner atoms, low-coordinated atoms on surface and atomic steps [J. K. Nørskov et al., *Nat. Chem.*, 1 (2009), 37] [Z. Li et al., *Nat. Catal.*, 2 (2019), 1107]. In this study, Turing structured PtNiNb nanosheet are synthesized with the morphology characteristics of labyrinthine networks composed by the irregular stripes. Such special morphology and substantial interfaces demonstrate

the fertile edge structures of Turing PtNiNb. We further conducted the XAFS experiments to reveal the traits of these edge structures. The PtNiNb nanosheet without Turing structure (the same composition as Turing PtNiNb) is prepared as the reference sample. The XAFS results show that the significant differences induced by the edge structures of Turing configuration might be the changes in coordination number. The Pt coordination number of Turing PtNiNb (8.1) is slightly higher than that of the PtNiNb without Turing structure (7.4). In contrast, Turing structuring contributed to the considerably reduction in the coordination number of Ni (9.1 to 6.7) and Nb (9.0 to 4.6) compared with the PtNiNb sample. These results demonstrate that the edge structures of Turing configuration, including the surface of the irregular stripes, interfaces and twin boundaries, have abundant low-coordinated atoms. These findings are also consistent with the features of stacking-fault-activated Ag nanoparticle reported previously, in which stacking fault leading to a low coordination number [Z. Li et al., *Nat. Catal.*, 2 (2019), 1107]. In addition, the reduced coordination number could be attributed to the significantly increased proportion of surface atoms on the Turing stripes (the coordination number of surface atoms are generally lower than that of the inner atoms). This conclusion can be further supported by the greatly increased electrochemical active surface area (Supplementary Table 6 of the revised manuscript) of Turing PtNiNb. Furthermore, we conducted supplementary experiments to investigate the electrocatalytic activity of PtNiNb nanosheet with and without Turing structure (the TEM images are shown in Response-Figure 7a and b). LSV curves show that the catalytic activity of Turing PtNiNb is much higher than that of the PtNiNb nanosheet without Turing structure (Response-Figure 7c). The overpotentials at 10 mA cm^{-2} Turing PtNiNb and PtNiNb nanosheet without Turing structure are 18.0 mV and 127.5 mV, respectively. These comparisons suggest that Turing structure presents significant structural advantages in alkaline HER. The significantly improved activity of PtNiNb catalyst is attributed to the Turing configuration, especially the edge structure and the crystalline defects (twins and stacking faults).

Nanotwins and stacking faults are two of the fundamental characteristics of the edge structure and Turing configuration in Turing PtNiNb. The effects of these crystalline defects on electrocatalytic activity are analysed by DFT calculations. According to the reviewer's advice, we have entirely re-constructed the calculation models based on the experimental composition of Turing PtNiNb ($\text{Pt}_{56.1}\text{Ni}_{33.5}\text{Nb}_{10.4}$). The sub-units of stacking faults highly resemble those of twins structure for an fcc structure. Therefore, the PtNiNb twin model with two parallel twin boundaries could be applied to reliably estimate the fundamental characteristic of multi-fold twins and stacking faults (Response-Figure 8a).

Crystallogometry analysis reveals that the introduction of twins configuration could result in partial symmetry breaking of a perfect fcc structure, and changes in orientation as well as interval of the atoms on sides of twin boundary. These crystalline features of twins-configuration are expected to induce significant changes in the local coordination environment of atoms on and near the twin boundary. DFT calculations show that the incorporation of twin configuration in PtNiNb twin slab yielded more active sites compared with the PtNiNb slab. For example, nine active sites of PtNiNb twin slab shows the ΔG_{H^*} values between ± 0.200 eV and four active sites have the ΔG_{H^*} values between ± 0.100 eV. As a contrast, PtNiNb slab without twin has only six active sites with the ΔG_{H^*} values between ± 0.200 eV and one active site with the ΔG_{H^*} values between ± 0.100 eV (Response-Figure 8b). Specifically, the atoms on and near the twin boundary (the top site of Ni on TB; the top site of Nb near TB; the hollow site of Pt-2 on TB; as shown in Supplementary Fig. 24 of the revised manuscript) contribute to the three of the most active sites in the PtNiNb twin. Hereafter, lattice strain was introduced in fcc PtNiNb slab. It is found that the compression strain leads to the interaction optimization between H intermediates and catalyst surface. The PtNiNb twin exhibits the optimized ΔG_{H^*} values of -0.012 eV and about 0.000 eV under 1% and 2% compression strain, respectively. Furthermore, the PtNiNb twin with 2% compressive strain shows a significantly increased density of active sites: eight active sites with ΔG_{H^*} values between ± 0.100 eV, which is 2-fold the PtNiNb twin without

lattice strain. In this case, the atoms on the twin boundary contribute seven considerably active sites with the ΔG_{H^*} values between ± 0.100 eV (the top sites of Pt and Nb on TB; the hollow sites of Pt-1, Pt-2 and Pt-3 on TB; the hollow site of Pt-4 near TB; the bridge site of Ni-Ni-1 near TB; as shown in Supplementary Fig. 24). These results illustrate that twin boundary could provide extra active sites, especially the Pt sites on twin boundary. Furthermore, calculation results show that Ni on twin boundary are stable site for the adsorption of H_2O and OH. The significantly reduced kinetics barrier of H_2O dissociation on the Ni site on twin boundary, which could accelerate the H_2 evolution and greatly improve the catalytic activity (Response-Figure 8c ,d and e).

The revisions corresponding to this comment are added in Page 14, Paragraph 2; the “**Theoretical calculation on Pt-Ni-Nb Turing configurations**” section (Page25 and Page26) of the revised manuscript.

Response-Figure 7 a, TEM image of PtNiNb without Turing structure. **b**, HRTEM image of PtNiNb without Turing structure. **c**, LSV curves of Turing PtNiNb and PtNiNb

without Turing structure in 1.0 M KOH. RHE, reversible hydrogen electrode. **d**, Tafel plots of Turing PtNiNb and PtNiNb without Turing structure.

Response-Figure 8 a, Top view and side view of PtNiNb twin calculation model, where the grey grid represents twins configuration and red dashed lines are twin boundaries. **b**, The hydrogen adsorption Gibbs free energy (ΔG_{H^*}) of bimetal PtNi, PtNiNb and PtNiNb twin models, the synergic effects of twin configuration and lattice strain on ΔG_{H^*} is investigated. **c**, Free-energy diagram for water dissociation on Pt, PtNiNb, PtNiNb twins and PtNiNb twins with 2% compressive strain. The dynamic atom-configurations for water dissociation on the facet of PtNiNb (**d**) and PtNiNb twins with 2% strain (**e**), including the H₂O adsorption, the transition state and the final state.

Reviewer #2 (Remarks to the Author): The manuscript by Lu et al. reported the superthin PtNiNb nanosheets with unique Turing stripes. The morphology regulation of Pt-based nanomaterials is crucial for the enhancement of atomic utilization rate, active site density and exposed specific active planes. In this work, the as-described Turing structure is quite interesting, whose spatiotemporal stationary pattern and concomitant high-density nanotwins/large lattice strain endow it with high intrinsic activity and stability. However, three significant problems were important for the integrity and persuasiveness of this work, but not well explained which let me reject it to be published in Nature Communications:

1. Why could the ordinary two-step magnetron sputtering followed by the chemical etching process result in the distinctive Turing structure? Is it affected by the Si sacrificing temple? These issues needs more supplemental evidences, contrast experiments and theoretical bases to expound the principle, feasibility and universality of the as-stated strategy.

Answer: Thanks for the reviewer's comments. The interpretation of this question consists of the following two parts. (1) The function and design purpose of Si layer; (2) The growth mechanism of Turing PtNiNb; (3) The universality of preparation strategy and Turing structure.

(1) The function and design purpose of Si layer

The Si layer was first deposited on the substrate and then the PtNiNb film was sputtered on the Si precursor layer. The etching of Si layer could release the PtNiNb film from the substrate. The Si-layer design can be attributed to the following two reasons:

First, the as-deposited metallic film on monocrystalline silicon and glass are very flat at both macroscopic and microscopic scale. The low specific surface area of flat metallic films can hardly lead to a significant improvement in catalytic activity [Z.J. Wang et al., *Adv. Mater.*, 32 (2020), e1906384]. In contrast, the exfoliated PtNiNb film by etching Si layer can be further ultrasonicated to form nanosheets with lateral sizes ranging from 100 to 500 nm. This kind of nanosheet-typed configurations could greatly

increase the electrochemical active surface area (Supplementary Table 6 of the revised manuscript). Here, we conducted supplementary experiments to verify the effects of sample forms on catalytic activity. We prepared three types of samples for comparison, including the catalyst ink of Turing PtNiNb (bimetal PtNi nanosheet, monometallic Pt 2D metal) dropped on carbon cloth, the Turing PtNiNb film (bimetal PtNi film, monometallic Pt film) deposited on carbon cloth and flat nickel plate. These PtNiNb, PtNi and Pt samples are prepared respectively using the same deposition method. As shown in the LSV curves (Response-Figure 9), the catalytic activity of monometallic Pt 2D metal dropped on carbon cloth is greatly higher than those of Pt films deposited on carbon cloth and flat nickel plate. The Turing PtNiNb samples and bimetal PtNi samples show the similar difference in catalytic performance: the catalytic performance of the nanosheet-typed samples is superior to the film-typed samples deposited on carbon cloth electrode and nickel electrode.

Response-Figure 9 The LSV curves of **a**, Turing PtNiNb, **b**, PtNi 2D and **c**, Pt 2D metal prepared via different methods.

Second, membrane-electrode-assembly is considered as one of the most competitive techniques for clean hydrogen production. We intended to research the performance of Turing catalysts and application potential in the anion-exchange-membrane water electrolyser. Membrane electrode and carbon supporting materials, as the gas diffusion layer, are frequently assembled in the device. The catalyst ink of Turing PtNiNb nanosheet can be sprayed on carbon cloth and carbon paper with controllable Pt loading mass. Therefore, the Turing PtNiNb can be compared with the commercial Pt/C (powder) with regard to the activity and stability of anion-exchange-membrane water electrolyser.

(2) The growth mechanism of Turing PtNiNb.

A.M. Turing elucidated the mechanism of the morphogenesis (spatiotemporal stationary patterns) by the reaction–diffusion theory, in which a pair of activator-inhibitor chemical substances react and diffuse with each other to yield Turing structure [A. M. Turing, *Philos. Trans. R. Soc. Lond. B Biol. Sci.*, 237 (1952), 37]. The necessary conditions for the generation of Turing structure are the competing positive and negative kinetic feedback between the activator-inhibitor pair. In addition, the inhibitor must have a larger diffusion coefficient than activator to promote the short-range activation and long-range inhibition phenomenon [A. Gierer et al., *Kybernetik*, 12 (1972), 30]. In an inorganic solid system, a nanoscale Turing bismuth monolayer was prepared on NbSe₂ surface by molecular beam epitaxy [Y. Fuseya et al., *Nat. Phys.*, 17 (2021), 1031]. The authors mathematically described the interactions between activator and inhibitor using the three-body potential of Bi and Se: elastic potential, adsorption potential and bond angle potential. This formation of Bi Turing monolayer was attributed to the small bond angle potential induced by the unstable Bi-structure/phase (Bi has ten potential solid phases under pressure). However, modelling the interactions in the complex PtNiNb-Si system is considerably difficult, because the multi-components of interactions potentials (film-substrate and film-film). To be honest, we

are still unable to provide a clear elucidation on the origin and underlying mechanism of the PtNiNb Turing structure. We are going to make hard efforts to find out the mechanism of this phenomenon.

(3) The universality of preparation strategy and Turing structure.

In the previous manuscript, we mainly focused on one Turing structure of the PtNiNb alloy. In order to verify the universality of the Turing structure and the as-reported preparation strategy, more Turing catalysts with different alloy composition should be demonstrated in this study. Therefore, we had conducted experiments in different alloy systems and demonstrated the successful achievement of Turing structure in Ir-Ni-Nb, Pd-Ni-Nb and Ag-Ni-Nb alloys using the same synthesis method of Turing PtNiNb (Response-Figure 10). The TEM images show the clear Turing structures of all the as-prepared Ir/Pd/Ag-Ni-Nb catalysts: labyrinthine pattern composed of irregular stripes (Response-Figure 10a, b and c). These Turing structures resemble the morphology characteristics of Turing PtNiNb. In addition, these Turing catalysts exhibited greatly enhanced HER performance with lower overpotentials compared with commercial 20 wt% Ir/C, Pd/C and Ag/C, respectively (Response-Figure 10). The successful preparation of Turing catalyst in different alloys significantly supports the generic significance of Turing structure.

Response-Figure 10 TEM images of **a**, Turing Ir-Ni-Nb nanosheet, **b**, Turing Pd-Ni-Nb nanosheet and **c**, Turing Ag-Ni-Nb nanosheet. LSV curves measured in 1.0 M KOH of **d**, Turing Ir-Ni-Nb and 20 wt% Ir/C, **e**, Turing Pd-Ni-Nb and 20 wt% Pd/C and **f**, Turing Ag-Ni-Nb and 20 wt% Ag/C. Comparisons for overpotential and Tafel slopes between **g**, Turing Ir-Ni-Nb and 20 wt% Ir/C, **h**, Turing Pd-Ni-Nb and 20 wt% Pd/C and **i**, Turing Ag-Ni-Nb and 20 wt% Ag/C.

2. The valence bond structure and atomic coordination information are missing.

Answer: Thanks for the reviewer's comments. We have additionally conducted the X-ray absorption spectroscopy experiments, and carefully discussed the valence bond structure as well as atomic coordination information. **The following content and Figures were added in the revised manuscript to reveal XAS results:**

The Turing PtNiNb presents a class of novel topological structure in nanometallic materials. We performed X-ray absorption near-edge structure (XANES) and extended X-ray absorption fine structure (EXAFS) experiments to reveal the electronic structure and coordination information of Turing structure. The Pt L₃-edge XANES spectra shows that the absorption thresholds and white line peaks of the tested samples Turing

PtNiNb ($\text{Pt}_{56.1}\text{Ni}_{33.5}\text{Nb}_{10.4}$), $\text{Pt}_{56.1}\text{Ni}_{33.5}\text{Nb}_{10.4}$ (PtNiNb, without Turing structure), bimetal $\text{Pt}_{63.7}\text{Ni}_{36.3}$ and Pt 2D metal are close to those of the Pt foil, suggest the Pt valence states are almost identical to zero (Response-Figure 11a below). The Fourier-transformed EXAFS spectra of Pt foil and Pt 2D metal shows the similar coordination environment, and the corresponding dominating peaks are assigned to Pt-Pt (~ 2.77 Å) coordination (Response-Figure 11d below). Ni doping results in the left shift of the dominating peak in the EXAFS spectra of binary PtNi sample, which is resolved as the Pt-Ni (2.62 Å) and Pt-Pt (2.69 Å) coordination. In contrast, Turing PtNiNb and PtNiNb have greatly different EXAFS spectra with significantly reduced amplitude of the atomic shell peaks. The fitting results of EXAFS spectrum reveal the major coordination of Pt-Pt (2.72 Å), Pt-Ni (2.59 Å) and Pt-Nb (2.67 Å) in Turing PtNiNb sample. The changes in peaks amplitude suggest the reduced coordination number. The fitting results show that the coordination number values of both Turing PtNiNb (8.1) and PtNiNb (7.4) are lower than those of bimetal PtNi (9.5), Pt 2D metal (9.5) as well as Pt foil (12, Supplementary Table 1 of the revised manuscript). Such pronounced reduction in coordination number is attributed to the small nanosheet size/thickness, fertile edge structures and substantial crystalline defects, including the nanotwins and stacking faults. The Pt coordination environment was further analysed by wavelet transform (WT) spectra at Pt L_3 -edge. The WT spectrum in Response-Figure 11g shows that the maximum peak intensity of the Pt foil is located at 10.05 Å⁻¹, which originates from the Pt-Pt bond. The maximum peak intensity of bimetal PtNi shifts to 7.35 Å⁻¹ due to the contribution of Pt-Ni bond (Response-Figure 11h). Furthermore, the Pt-M paths of Turing PtNiNb shifts to a lower k space, in sharp contrast to the Pt-Pt coordination in Pt foil (Response-Figure 11i).

Response-Figure 11b shows the normalized Ni K-edge XANES spectrum. A clear pre-edge around 8333 eV was detected, suggesting the dipole-forbidden, quadrupole-allowed transition ($1s$ to $3d$) [N. Becknell et al., J. Am. Chem. Soc., 137 (2015), 15817]. The absorption edge of the XANES spectrum was applied to evaluate the Ni valence. Turing PtNiNb presents a higher adsorption threshold and a greater intensity of white

line compared with the Ni foil, indicating the variations in electronic structure. Ni in Turing PtNiNb is speculated to be positively charged and parts of nickel species are oxidation states. The Ni valence states were further confirmed with the Ni 2p high-resolution X-ray photoemission spectra (XPS) showing the characteristic peaks of Ni²⁺ (Supplementary Fig. 8 of the revised manuscript). The Fourier transformed EXAFS spectra of bimetal PtNi sample shows a dominating peak at around 2.2 Å, which originates from the first shell of Ni-Ni coordination (Response-Figure 11e). The coordination environments of bimetal PtNi highly resemble those of Ni foil and NiNb nanosheet. Taking the bimetal PtNi as the reference, Nb doping contribute to the left shift in the dominating peaks of PtNiNb. Fitting results on the EXAFS spectrum of PtNiNb illustrate that the major coordination is Ni-Ni and the second peak locating around 2.65 Å can be attributed to the Ni-Nb coordination. By contrast, Turing PtNiNb present two distinct scattering paths in the EXAFS spectrum. The first peak at ~ 1.6 Å is assigned to Ni-O coordination, while the second peak at ~ 2.8 Å is speculated to originate from the second coordination shell of the Ni-Ni scattering path. The Ni-Pt coordination with a small coordination number of 0.6 was also resolved from the fitting of EXAFS spectrum (Supplementary Table 2 of the revised manuscript).

We also analyzed the XANES and EXAFS spectra of these sample at Nb K-edge. Turing PtNiNb, PtNiNb (without Turing structure) and NiNb samples show higher near-edge absorption energies and larger intensity of white line peaks than those of Nb foil, suggesting the higher Nb valence states in these three samples (Response-Figure 11c). The Fourier transformed EXAFS spectra of Turing PtNiNb and NiNb depicts the peak components between 1.2 Å and 1.9 Å as a result of Nb-O scattering pair, while this peak is relatively weak in PtNiNb sample (Response-Figure 11f). Furthermore, fitting calculations on the EXAFS spectrum of Turing PtNiNb resolved the Nb-Nb, Nb-Pt, Nb-Ni as well as Nb-O coordination. Turing structuring induced a significant reduction in coordination numbers. The coordination number of Nb in Turing PtNiNb (4.6) is even smaller than that of amorphous NiNb nanosheet (7.0) which represents a kind of low-coordination structure (Supplementary Table 3 of the revised manuscript).

The greatly reduced coordination number of Ni and Nb elements in Turing PtNiNb indicate that the edge structures of Turing configuration, including the surface of the irregular stripes, interfaces and twin boundaries, have abundant low-coordinated atoms (Supplementary Table 2 and 3 of the revised manuscript). These findings are also consistent with the features of stacking-fault-activated Ag nanoparticle reported previously, in which stacking fault leading to a low coordination number [Z. Li et al., Nat. Catal., 2 (2019), 1107]. In addition, the reduced coordination number could be attributed to the significantly increased proportion of surface atoms on the Turing stripes (the coordination number of surface atoms are generally lower than that of the inner atoms). This conclusion can be further supported by the greatly increased electrochemical active surface area of Turing PtNiNb (Supplementary Table 6 of the revised manuscript).

The revisions corresponding to this comment are added in Page 12, Paragraph 2; Page 13 and Page 14; Fig. 3 of the revised manuscript.

Response-Figure 11 Electronic structure analysis. XANES spectra collected at Pt L₃-edge (a), Ni K-edge (b) and Nb K-edge (c). Fourier transformed EXAFS spectra at Pt L₃-edge (d), Ni K-edge (e) and Nb K-edge (f). Wavelet transforms analysis of the Pt L₃-edge for Pt foil (g), PtNi 2D metal (h) and Turing PtNiNb (i).

3. The theoretical modes used in this work is in consistent with the actual structure. The main product is Pt_{56.1}Ni_{33.5}Nb_{10.4} with the material phase of fcc Pt/PtNi. But the DFT model uses a fcc Pt₃Ni. The effect of Nb element is disregarded and the elemental composition is totally different. That is definitively wrong. With the reasonable model, the crystal structure of pristine crystal plane, and the planes with incorporated twins and strain should be exhibited, in order to explain how the twins and strain are introduced. Furthermore, 1) what kind of strain is exerted? Tensile or compressive strain? 2) Why did the Pt 2D metal use the Pt(111) as calculation model? 3) The electronic structure calculations need to be supplemented.

Answer: Thanks for the reviewer's comments. We have entirely re-constructed the calculation models according to the experimental composition of Turing PtNiNb (Pt_{56.1}Ni_{33.5}Nb_{10.4}). In the revised theoretical calculations, the atomic proportions of the fcc PtNiNb and fcc PtNiNb twin slab models are the same: Pt_{55.6}Ni_{33.3}Nb_{11.1}, close to the experimental composition of Pt_{56.1}Ni_{33.5}Nb_{10.4}. The subtle composition difference originates from the feature of atomic proportion in Pt_{56.1}Ni_{33.5}Nb_{10.4}, where the same atomic composition is unavailable in the models with finite atoms. The PtNiNb twin model is exhibited in Response-Figure 12a below. The responses to the sub-questions are listed as follows:

1) what kind of strain is exerted? Tensile or compressive strain?

Answer: In the previous calculations, the compressive stain was exerted on the slab models to calculate the free energy and energy barrier. We have stated the strain types exerted on the atomic model in our revised manuscript. And the effects of both

compressive and tensile strains on catalytic properties are investigated and discussed in the revised calculations.

2) Why did the Pt 2D metal use the Pt(111) as calculation model?

Answer: This question is interpreted from the following two aspects: a. As for an fcc structure, the surface energies of the low-index facets are estimated as $\gamma_{\{100\}} = 4(\varepsilon/a^2)$, $\gamma_{\{110\}} = 4.24(\varepsilon/a^2)$ and $\gamma_{\{111\}} = 3.36(\varepsilon/a^2)$, where γ is the surface energy, ε is the bond strength and a is the lattice constant [Y.N. Xia et al., *Angew. Chem. Int. Ed.*, 48 (2009), 60]. Generally, the {111} facets are expected to have lower surface energies than {100} and {110} facets, suggesting {111} facets are relatively stable crystallographic surfaces for an fcc monometal; b. Our calculation has referred to the methods and reference samples employed in literatures. Pt (111) model is always considered as the reliable benchmark to evaluate the catalytic activity of Pt catalysts and be compared with the new Pt-based catalysts. For example, the studies on Pt/Ru-Pt catalysts, Pt/PtTe₂ nanosheets, Pt/Pt-Ni nanowire and high-entropy Pt nanosponge, have calculated the Pt (111) properties to estimate the catalytic activity of Pt materials and further applied the Pt (111) model as the control sample [S. Zhu et al., *Nat. Cat.*, 4 (2021), 711] [P. Wang et al., *Nat. Commun.*, 8 (2017), 14580] [Z. Jia et al., *Adv. Funct.*, (2021), 2101586].

3) The electronic structure calculations need to be supplemented.

Answer: We has supplemented the electronic structure calculations. The relations between electronic structures and catalytic activity were carefully discussed in the revised manuscript.

We re-conducted the theoretical calculations using the new atomic models, which aims to address the above concerns of the reviewer. We also re-discussed the calculation results, and the following content is added to the revised manuscript as the “**Theoretical calculation on Pt-Ni-Nb Turing configurations**” section.

The above analysis reveals Turing PtNiNb with the major crystallographic features

of multiple nanotwins and lattice strain. We performed theoretical calculations to disclose the mechanism of enhanced HER activity induced by Turing structuring. Experimental data shows that the multi-fold nanotwins in Turing PtNiNb are mainly enclosed by {111} and {200} atomic planes with [110] zone axis. We cleaved the fcc Pt₃Ni (110) layers and substituted Pt/Ni atoms to construct the four-atomic-layer fcc PtNi, fcc PtNiNb and fcc PtNiNb twin slab models. The fcc PtNi slab has a Pt : Ni ratio of 62.5 : 37.5, which is almost consistent with the composition of bimetal Pt_{62.6}Ni_{37.4} sample. The atomic proportions of the fcc PtNiNb and fcc PtNiNb twin slab models are the same: Pt_{55.6}Ni_{33.3}Nb_{11.1}, close to the experimental composition of Turing PtNiNb (Pt_{56.1}Ni_{33.5}Nb_{10.4}). Extensive structure optimization calculations were carried out to determine the stable slab configurations, and the results of fcc PtNiNb twin slab model is shown in Response-Figure 12a below.

The hydrogen adsorption Gibbs free energy (ΔG_{H^*}), a widely used descriptor for HER activity, was calculated to study the interaction between H* and catalysts [J.K. Nørskov et al., J. Electrochem. Soc., 152 (2005), J23]. The Nb doping and the incorporation of twin-configuration in fcc PtNiNb slab model greatly change the local atomic configurations (Supplementary Fig. 23 and 24 of the revised manuscript). Response-Figure 12b shows the summary of calculated values of ΔG_{H^*} . The bimetal PtNi slab presents an optimal ΔG_{H^*} (0.046 eV, the bridge site of Ni-Ni) which is weaker than the ΔG_{H^*} of Pt (111) slab (-0.266 eV), revealing the positive contribution of Ni doping to catalytic activity. The Nb doping could greatly enrich the coordination types and optimize the ΔG_{H^*} of PtNiNb slab to -0.010 eV (the hollow site of Pt-2 shown in

Supplementary Fig. 23 of the revised manuscript). Moreover, five other active sites of the PtNiNb slab also shows the ΔG_{H^*} values between ± 0.200 eV (the top site of Nb, the bridge sites of Pt-Nb and Ni-Ni, the hollow sites of Pt-1 and Ni, as shown in Supplementary Fig. 23 of the revised manuscript), suggesting the beneficial effects of Nb in improving the H^* adsorption/desorption on catalyst surface. As for the PtNiNb twin slab, calculation results show the optimized ΔG_{H^*} of 0.047 eV which is larger than the smallest ΔG_{H^*} of the PtNiNb slab. Despite of the inferior values of ΔG_{H^*} , the incorporation of twin configuration in PtNiNb twin slab has yielded more active sites compared with the PtNiNb slab. For example, nine active sites of PtNiNb twin slab show the ΔG_{H^*} values between ± 0.200 eV and four active sites have the ΔG_{H^*} values between ± 0.100 eV. Specifically, the atoms on and near the twin boundary (the top site of Ni on TB; the top site of Nb near TB; the hollow site of Pt-2 on TB; as shown Supplementary Fig. 24 of the revised manuscript) contribute to the three of the most active sites. Twin configuration could result in partial symmetry breaking of a perfect fcc structure, and changes in orientation as well as interval of the atoms on sides of twin boundary. These crystalline features of twins are expected to induce significant changes in the local atomic coordination. The greatly increased number of active sites illustrate that the variation in atomic coordination induced by twin boundary can enormously improve catalytic activity. Hereafter, lattice strain was introduced in fcc PtNiNb slab. The statistical results on ΔG_{H^*} under compressive strain and tensile strain are shown in Response-Figure 12b. It is found that the compressive strain leads to the interaction optimization between H intermediates and catalyst surface. The PtNiNb twin exhibits

the optimized ΔG_{H^*} values of -0.012 eV and about 0.000 eV under 1% and 2% compressive strain, respectively. This ΔG_{H^*} value (0.000 eV) of PtNiNb twin with 2% compressive strain (a hollow site of Pt on twin boundary) is close to the optimum value of ΔG_{H^*} for HER (0 eV), indicating the ideal moderate interaction between intermediates and active sites [J.K. Nørskov et al., *J. Electrochem. Soc.*, 152 (2005), J23]. According to Sabatier principle, this interaction, neither strong nor weak, is ideal for the adsorption/desorption of reactive participants [Z.W. She et al., *Science*, 355 (2017), eaad4998] This ideal interaction assures the desorption of H^* to form H_2 facily in the Tafel step. The weaker adsorption of hydrogen on Turing PtNiNb was further confirmed by CV measurements with the left shift of underpotentially deposited hydrogen (H_{upd}) peak from 0.236 V vs. RHE for Pt/C to 0.198 V vs. RHE for Turing PtNiNb (Supplementary Fig. 25 of the revised manuscript) [X. Li et al., *Nat. Commun.* 9 (2018), 4958]. Furthermore, the PtNiNb twin with 2% compressive strain shows a significantly increased density of active sites: eight active sites with ΔG_{H^*} values between ± 0.200 eV, which is 2-fold the PtNiNb twin without lattice strain. In this case, the atoms on the twin boundary contribute seven considerably active sites with the ΔG_{H^*} values between ± 0.100 eV (the top sites of Pt and Nb on TB; the hollow sites of Pt-1, Pt-2 and Pt-3 on TB; the hollow site of Pt-4 near TB; the bridge site of Ni-Ni-1 near TB; as shown in Supplementary Fig. 24 of the revised manuscript). These results illustrate that twin boundary could provide extra active sites. In contrast, tensile strain results in the general reduction in ΔG_{H^*} of PtNiNb twin. The reduction in ΔG_{H^*} suggests the enhanced adsorption of reaction intermediates on the catalyst surface, thus slowing

the H^* conversion and H_2 evolution.

The optimized interaction between reaction intermediates and catalyst surface may be attributed to the changes in electronic structure. The projected density of states (PDOS) of d orbital were further analysed and the results are shown in Response-Figure 12. Ni has a higher density near the Fermi level compared with Pt, and this trait indicates that Ni could donate more free electrons to near atoms during HER. Nb shows a broad band, but relative low density of occupation states. Nb doping in PtNi results in downshift of d -band centre from -2.82 eV to -2.94 eV for Pt sites (Response-Figure 12c and d). The downshift of d -band centre indicates that the transferred electronic would fill d -band, i.e., the antibonding electronic states [B. Hammer et al., *Nature* 376 (1995), 238] [J. K. Norskov et al., *Proc. Natl. Acad. Sci. U. S. A.* 108 (2011), 937]. Therefore, Nb-doping could suppress the over-binding of H^* on catalyst surface and weakening the H^* adsorption, improving ΔG_{H^*} to optimal value. This result is in accordance with the optimized ΔG_{H^*} in comparison with PtNiNb (-0.010 eV) and PtNi (0.046 eV). The effects of twin configuration on electronic structure were also investigated. And the results show that the introduction of twin configuration in PtNiNb lead to a upshift of d -band center from -2.94 eV (PtNiNb without twin) to -2.86 eV (Response-Figure 12e). Such changes in d -band centre provide the theoretical interpretation to the ΔG_{H^*} deterioration from -0.010 eV (PtNiNb without twin) to 0.047 eV (PtNiNb twin). Nevertheless, the d -band center of the atoms on twin boundary was downshifted by 0.13 eV due to the introduction of twin configuration. The twin boundary may thus provide more active sites for the catalysis. Furthermore, the shift of d -band center can

be tailored by lattice strain. As shown in Response-Figure 12f, the *d*-band centers of PtNiNb twin and Pt sites are monotonously shifted towards negative energy direction by increasing the compressive strain. These results are consistent with the findings of ΔG_{H^*} calculations.

In alkaline HER, proton generally originates from the dissociation of water and this reaction is always considered as the rate-determining step. The experimental results suggest that the HER on Turing PtNiNb surface follows the Volmer-Tafel pathway. The transition states (TS) were searched along the corresponding reaction pathway and the reaction barriers are calculated. We first calculated the possible adsorption configuration of H₂O and OH. It's found that H₂O is preferentially adsorbed on Ni and Nb sites, but OH could be more stably adsorbed on Ni sites. The initial state and final state are therefore designed with H₂O-adsorbed on Ni sites, and OH-adsorbed on Ni sites with H-adsorbed on Pt sites, respectively. Both the PtNiNb and PtNiNb twin are calculated using the similar reaction pathway (Response-Figure 12h and i). The water dissociation on PtNiNb twin were proceeded on twin boundary due to the stable adsorption configuration. Calculation results demonstrated that the kinetic barriers of PtNiNb twin slab for H₂O dissociation (0.7042 eV) was lower than those of PtNiNi slab (0.791 eV) and Pt (111) slab (Response-Figure 12g). Besides, lattice strain facilitated water dissociation and further reduced the kinetic barrier to 0.605 eV. The lower dissociation barrier of water could markedly assist the proton production from water and facilitate the subsequent H₂ evolution. Therefore, twin configuration and lattice strain can significantly contribute to the high catalytic activity of PtNiNb in alkaline

HER.

Response-Figure 12. **a**, Top view and side view of PtNiNb twin calculation model, where the grey grid represents twins configuration and red dashed lines are twin boundaries. **b**, The hydrogen adsorption Gibbs free energy (ΔG_{H^*}) of bimetal PtNi, PtNiNb and PtNiNb twin models, the synergic effects of twin configuration and lattice strain on ΔG_{H^*} are investigated. The projected electronic density of states for d orbital of PtNi (c), PtNiNb (d) and PtNiNb twin (e), including the PDOS diagram of free H_2O molecule. **f**, The d -band centres of PtNiNb twins as a function of tensile strains (positive) and compressive strains (negative). **g**, Free-energy diagram for water dissociation on Pt, PtNiNb, PtNiNb twins and PtNiNb twins with 2% compressive strain. The dynamic atom-configurations for water dissociation on the facet of PtNiNb (h) and PtNiNb twins with 2% strain (i), including the H_2O adsorption, the transition state and the final state.

The above three problems make the structure-activity unconvincing, thus, I cannot support its publication in Nature Communications.

Answer: Thanks for the reviewer's comments. These insightful suggestions certainly help us to better understand this work. We believe that the extensive experiments and calculations supplemented according to these suggestions would well address the reviewers' concerns and strengthen the novelty of this work. We anticipate that the reviewer could re-evaluate this work and consider its publication sincerely.

Some small issues are listed as follows:

a) Why was the "PtNiNb nanosheets" abbreviated to "Turing PtNi"? The Nb is totally ignored. And the actual bimetal PtNi contrast sample is missing, which is very important in this work, especially for the electrochemical evaluation.

Answer: Thanks for the reviewer's comments. Platinum and nickel are the major elements in the $\text{Pt}_{56.1}\text{Ni}_{33.5}\text{Nb}_{10.4}$ metallic catalyst. In the previous manuscript, we intended to underline the important contribution of nanotwins and lattice strain, which are the intrinsic characteristics of Turing structure. Therefore, niobium element and its effects on structure as well as catalytic properties were erroneously ignored. However, the reviewers' comment and the supplemental experiments conducted to address the reviewers' concerns make the effects of niobium doping much clearer and help us to better understand the significance of niobium in the high-performance Turing catalyst. Therefore, we correct this abbreviation and PtNiNb nanosheet is denoted as Turing PtNiNb in the revised manuscript.

In addition, we synthesized the bimetal $\text{Pt}_{63.7}\text{Ni}_{36.3}$ catalyst using the same preparation method of Turing PtNiNb. The microstructure and catalytic property of bimetal $\text{Pt}_{63.7}\text{Ni}_{36.3}$ catalyst was further studied as one of the benchmarks. TEM images show that bimetal $\text{Pt}_{63.7}\text{Ni}_{36.3}$ catalyst are prepared as nanosheets without any traits of Turing structure (Response-Figure 13). The $\text{Pt}_{63.7}\text{Ni}_{36.3}$ nanosheets are composed of continuous nanosized crystals with random orientations, corresponding to the classical morphology of nanometallic materials. Structural difference between $\text{Pt}_{63.7}\text{Ni}_{36.3}$

catalyst and Turing PtNiNb catalyst suggests the importance of Nb addition on the formation of Turing structure. In addition, Response-Figure 14 presents the LSV curves and Tafel slopes of bimetal Pt_{63.7}Ni_{36.3} together with Turing PtNiNb, Pt 2D metal as well as commercial Pt/C. The catalytic activity of bimetal Pt_{63.7}Ni_{36.3} is lower than that of Turing PtNiNb, but higher than Pt 2D metal and Pt/C. For example, the overpotential at 10 mA cm⁻² increase in the order Turing PtNiNb (18.0 mV), bimetal Pt_{63.7}Ni_{36.3} (27.2 mV), Pt 2D metal (33.6 mV) and Pt/C (51.0 mV). The Tafel slopes of Turing PtNiNb, bimetal Pt_{63.7}Ni_{36.3}, Pt 2D metal and Pt/C are 32.7 mV dec⁻¹, 35.3 mV dec⁻¹, 35.6 mV dec⁻¹ and 36.0 mV dec⁻¹, respectively (Response-Figure 14b). Furthermore, catalytic stability of binary Pt_{63.7}Ni_{36.3} catalyst was also evaluated in 1.0 M KOH electrolyte. Bimetal Pt_{63.7}Ni_{36.3} catalyst shows a large overpotential change of 170.5 mV in chrono-potentiometric test and pronounced decline of current density in the accelerated durability test (Response-Figure 14c and d). Turing PtNiNb shows the highest catalytic stability among the test samples. While the catalytic stability of bimetal Pt_{63.7}Ni_{36.3} is better than those of Pt 2D metal (187 mV) and Pt/C. These results illustrate the crucial role of niobium doping in structure modulation and activity improvement of Pt-Ni alloys. The experimental findings on bimetal Pt_{63.7}Ni_{36.3} can definitely support the structure and activity novelty of Turing PtNiNb.

The experimental data of bimetal PtNi catalyst has been added in Fig. 4 of the revised manuscript and we also revised the corresponding results description in the revised manuscript (Page 15 to Page 19).

Response-Figure 13. **a.** The TEM image of bimetal PtNi nanosheet. **d.** The HRTEM image of bimetal PtNi nanosheet.

Response-Figure 14. **a,** LSV curves of Turing PtNiNb, Pt 2D metal, PtNi 2D metal and Pt/C in 1.0 M KOH. RHE, reversible hydrogen electrode. **b,** Tafel plots of Turing PtNiNb, Pt 2D metal, PtNi 2D metal and Pt/C. **c,** Long-term stability test of Turing PtNi, Pt 2D metal, PtNi 2D metal and Pt/C under a large current density of 200 mA cm^{-2} in 1.0 M KOH. The inset is the HAADF-STEM image of Turing PtNi after the stability test. **d,** Variation of current density during accelerated durability test. Columns refer to values at 60 mV recorded in LSV curves prior to testing, and after 10,000 (10k).

b) Was the synthesis method for the contrastive Pt and NiNb (as well as the missing PtNi) the same as the main sample? Did the Turing structure also exist in these samples?

Answer: Thanks for the reviewer's comments. The contrastive Pt, NiNb and bimetal Pt_{63.7}Ni_{36.3} samples were synthesized using the same preparation method of Turing PtNiNb. However, the features of Turing structures were not found in these samples. As shown in Response-Figure 15 and Response-Figure 16, the as-synthesized Pt and bimetal Pt_{63.7}Ni_{36.3} nanosheets consist of densely interconnected nanocrystals with random orientations, which corresponds to the classical morphology of nanometallic materials. TEM images (Response-Figure 15c and d) of NiNb show the fully amorphous structure and diffused rings in the selected area electron diffraction patterns.

Response-Figure 15 a, b, TEM images of monometallic Pt 2D metal. Inset is the corresponding SAED patterns showing the sharp diffraction rings. The TEM results suggest that the monometallic Pt 2D metal is composed of nano-crystals with the grain-size smaller than 10 nm. **c**, TEM image of the NiNb 2D metal. **d**, HRTEM image of the NiNb 2D metal. Inset shows the corresponding SAED patterns. The maze-like patterns of atomic arrangement and the diffraction halos confirm the fully amorphous structure of the as-prepared NiNb 2D metal.

Response-Figure 16 a. The TEM image of bimetal PtNi nanosheet. **d.** The HRTEM image of bimetal PtNi nanosheet.

c) Electrochemical impedance spectroscopy should be supplemented to offer the solution resistance and charge transfer resistance.

Answer: Thanks for the reviewer's comments. We conducted supplemental experiments to analyse the impedance of Turing PtNiNb, bimetal PtNi, Pt 2D metal and NiNb nanosheet. The electrochemical impedance spectroscopies were tested at an overpotential of 50 mV with an amplitude of 5 mV, covering a frequency range of 10^{-1} Hz to 10^5 Hz. The Nyquist diagrams show the minimum magnitude of the semi-circle of Turing PtNiNb among all the tested samples (Response-Figure 17). Fitting results based on the equivalent circuit reveal that the charge transfer resistance of Turing PtNiNb (5.4 ohm) is lower than those of bimetal PtNi (6.8 ohm), Pt 2D metal (28.2 ohm) and NiNb nanosheet (46.5 ohm), which could significantly enhance the reaction kinetics and support the excellent HER activity of the Turing PtNiNb. The solution resistances of these tested samples are listed in Response-Table 1.

The revisions corresponding to this comment are added in Page 17, Paragraph 1; Supplementary Fig. 13 and Supplementary Table 5 of the revised manuscript.

Response-Figure 17. Nyquist plots collected at an overpotential of 50 mV for Turing PtNiNb, PtNi 2D metal, Pt 2D metal and NiNb 2D metal.

Response-Table 1 Solution resistances and charge transfer resistances corresponding to samples in Fig. based on fitting results.

Samples	Solution resistance (ohm)	Charge transfer resistance (ohm)
Turing PtNiNb	6.8	5.4
PtNi 2D metal	8.9	6.8
Pt 2D metal	14.3	28.2
NiNb 2D metal	12.0	46.5

d) Could the chemical etching process (immersed in 2 M KOH for 60 min) totally remove the Si layer?

Answer: Thanks for the reviewer's comments. Si content remaining in the as-synthesized samples was tested by ICP-MS and XPS. Si element can be hardly detected by the ICP-MS experiments in Turing PtNiNb, bimetal PtNi, Pt 2D metal and NiNb 2D metal samples. All these samples are synthesized through the as-stated etching process. Besides, no characteristic peaks of Si element can be found in the high-resolution XPS

spectra (Response-Figure 18). These experimental results suggest that Si element might be almost completely removed from the samples by this synthesis method.

Response-Figure 18 High resolution XPS spectra for Si 2p in **a**, Turing PtNiNb, **b**, PtNi 2D metal **c**, Pt 2D metal and **d**, NiNb 2D metal.

e) The catalyst ink is more suitable for the glassy carbon electrode measurement. Within the low current density range in this work, the glassy carbon electrode is not restricted by evident mass transfer problem, and can better reflect the intrinsic activity. Even with the carbon cloth electrode, the actual reaction area and the loading were not provided as well.

Answer: Thanks for the reviewer providing this crucial advice to strengthen our study. We carried out additional experiments using the glassy carbon electrode to investigate the intrinsic activity. The nanosheet catalyst in ethanol was difficult to form the intact and smooth ink film on the glassy carbon electrode, therefore the catalysts ink for glassy carbon electrode (GCE) were prepared with a mixture of carbon black (details in the method, Pt mass loading is about 10 μg). LSV curves measured on GCE shows Turing PtNiNb possesses the best performance with the lowest overpotential at 10 mA cm^{-2} of 23.7 mV, which is lower than the contrastive samples (PtNi 2D metal: 35.3 mV, Pt 2D metal: 43.2 mV, Pt/C: 53.4mV and NiNb 2D metal: 498.9 mV,

Response-Figure 19a). Moreover, Response-Figure 19d illustrates that Turing PtNiNb has a greatly higher mass activity of $3581.8 \text{ mA mg}^{-1}_{\text{Pt}}$ with comparison of PtNi 2D metal ($961.0 \text{ mA mg}^{-1}_{\text{Pt}}$), Pt 2D metal ($953.9 \text{ mA mg}^{-1}_{\text{Pt}}$) and Pt/C ($231.7 \text{ mA mg}^{-1}_{\text{Pt}}$). Tafel slopes of Turing PtNiNb, PtNi 2D metal, Pt 2D metal and Pt/C were all close to 29.9 mV dec^{-1} (Response-Figure 19c), indicating that the Tafel step ($2\text{M-H}^* \rightleftharpoons \text{H}_2 + 2\text{M}$) is the rate-determining step and the HER processed on Volmer-Tafel mechanism. These findings are consistent with the previous electrocatalytic results conducted on the carbon cloth electrodes.

Membrane-electrode-assembly is considered as one of the most competitive techniques for clean hydrogen production. In the previous manuscript, carbon cloth was applied as the supporting material, because we intended to research the catalytic performance of Turing catalysts and application potential in a complete cell configuration, i.e., the anion-exchange-membrane water electrolyser. Membrane electrode and carbon supporting materials, as the gas diffusion layer, are frequently assembled in the device. Meanwhile, some studies demonstrated the electrolyser properties by spraying nanocatalysts on the carbon cloth surface. Therefore, the carbon cloth electrode might be more reliable to evaluate the performance of Turing catalyst in the membrane-electrode-assembly. Considering the consistent results between carbon cloth electrode and glassy carbon electrode, we still present the results of carbon cloth electrodes and add the findings of glassy carbon electrode in the revised manuscript.

The revisions corresponding to this comment are added in Page 16, Paragraph 2; Supplementary Fig. 12 and Supplementary Table 4 of the revised manuscript.

Response-Figure 19 a, LSV curves of Turing PtNiNb, Pt 2D metal, PtNi 2D metal, Pt/C and amorphous NiNb 2D metal in 1.0 M KOH, with 85% resistance compensation. **b**, LSV curves without resistance compensation. **c**, Tafel plots of Turing PtNiNb, Pt 2D metal, PtNi 2D metal and Pt/C. **d**, Comparison for overpotential at 10 mA cm^{-2} and mass activity at 100 mV (vs. RHE) between samples in (a).

Reviewer #3 (Remarks to the Author): The authors report Turing structured PtNiNb electrocatalysts for hydrogen evolution reaction in an alkaline medium. The work looks like the authors have endeavoured to organize this manuscript from the good level of completion of this article (e.g., in-depth analysis for physical characterizations, performance evaluation, DFT, and post-testing studies). Especially the structural/topological studies on PtNiNb materials provide an in-depth understanding of the materials for lattice distortion, defects, twinning, or 3D architectures. I can agree that the material preparation & corresponding physical analyses are comprehensive in this study but the major claims and the structural novelty are less studied or not well-supported by experimental evidence or computational approaches. Some major comments that should be addressed are listed as follows to consider this article publishing in Nature Communications:

1. The catalyst named PtNi is Pt_{56.1}Ni_{33.5}Nb_{10.4} from ICP-OES analysis. What is the role of Ni & Nb atoms for this structural aspect and the electrochemical role?

Answer: Thanks for the reviewer's comments. We have conducted supplementary experiments and theoretical calculations to analyse the effects of Ni- and Nb-doping on structure and catalytic activity. It's noted that reviewer #1 and reviewer #2 raised similar questions on the doping effects. Therefore, we summarized these corresponding responses. The interpretation of this question consists of the following three parts: (1) Structural aspect; (2) Catalytic performance; (3) Theoretical analysis.

(1) Structural aspect

We synthesized the bimetal Pt_{63.7}Ni_{36.3} catalyst using the same preparation method of Turing PtNiNb and Pt nanosheet. Binary Pt_{63.7}Ni_{36.3} catalyst (denoted as PtNi 2D metal) has a Pt-Ni ratio of 1.75 that is close to the ratio of Pt to Ni (1.67) in Turing PtNiNb (Pt_{56.1}Ni_{33.5}Nb_{10.4}). The microstructure and catalytic property of bimetal Pt_{63.7}Ni_{36.3} catalyst was further studied as one of the benchmarks. TEM images show that bimetal Pt_{63.7}Ni_{36.3} catalyst is prepared as nanosheets without any traits of Turing structure (Response-Figure 20c and d). The Pt_{63.7}Ni_{36.3} nanosheets are composed of continuous nanosized crystals with random orientations, corresponding to the classical

morphology of nanometallic materials. This kind of classical morphology of nanometallic materials is similar to the microstructures of monometallic Pt 2D metal (Response-Figure 20a) and is sharply contrasted to the Turing structure (Response-Figure 20b). Binary Pt-Ni alloys are widely reported active electrocatalysts and are generally prepared as low-dimensional materials with an fcc structure, such as the Pt_{1.5}Ni as well as Pt₃Ni catalysts [X. Tian et al., *Science*, 366 (2019), 850] [P. Wang et al., *Nat. Commun.*, 8 (2017), 14580]. We did not find any special morphology features (Turing structure or amorphous phase) in the as-prepared Pt_{63.7}Ni_{36.3} nanosheet and the reported binary Pt-Ni catalyst, suggesting that Ni doping would not lead to significant structural changes. In contrast, the niobium-deletion in Pt-Ni alloys would lead to the breakdown of Turing structure and the enormous structure difference between Pt_{63.7}Ni_{36.3} catalyst and Turing PtNiNb catalyst. These findings underline the important role of niobium to induce the formation of Turing structure.

Response-Figure 20 a. TEM images of monometallic Pt 2D metal. b. TEM images of Turing PtNiNb. c. TEM images of bimetal PtNi. d. HRTEM image of bimetal PtNi.

(2) Catalytic performance

Electrocatalytic experiments were conducted to explore the effects of Ni and Nb on catalytic activity. Response-Figure 21 presents the LSV curves and Tafel slope of bimetal Pt_{63.7}Ni_{36.3} together with Turing PtNiNb and Pt 2D metal. The catalytic activity of bimetal Pt_{63.7}Ni_{36.3} is lower than that of Turing PtNiNb, but higher than that of Pt 2D metal. For example, the overpotential at 10 mA cm⁻² increase in the order Turing PtNiNb (18.0 mV), bimetal Pt_{63.7}Ni_{36.3} (27.2 mV) and Pt 2D metal (33.6 mV). The Tafel slopes of Turing PtNiNb, bimetal Pt_{63.7}Ni_{36.3} as well as Pt 2D metal are 32.7 mV dec⁻¹, 35.3 mV dec⁻¹ and 35.6 mV dec⁻¹, respectively. These results illustrate that Ni- and Nb-doping can significantly improve the alkaline HER activity of the Pt/PtNi catalysts.

Response-Figure 21 a, LSV curves of Turing PtNiNb, PtNi 2D metal and Pt 2D metal in 1.0 M KOH. RHE, reversible hydrogen electrode. **b**, Tafel plots of Turing PtNiNb, PtNi 2D metal and Pt 2D metal.

(3) Theoretical analysis

The mechanism of the high catalytic activity induced by Ni- and Nb-doping was further analysed by theoretical calculations. According to the below comments 2 of reviewer #3, we have re-constructed the atomic models and entirely re-conducted the DFT calculations. The Ni- and Nb-doping effects have been carefully investigated. And the details of discussion could be found in the below response to the comments 2 of reviewer #3. Here, we make a concise summary for reviewer's consideration: the Nb introduction in PtNi cause a downshift of *d*-band centre from -2.82 eV (PtNi) to -2.94 eV (PtNiNb) for Pt sites. Such downshift of *d*-band centre indicates that the antibonding

states were further filled which could weaken the H adsorption [B. Hammer et al., *Nature* 376 (1995), 238] [J. K. Norskov et al., *Proc. Natl. Acad. Sci. U. S. A.* 108 (2011), 937]. This result is consistent with the optimization in the hydrogen adsorption Gibbs free energy (ΔG_{H^*}) of PtNiNb (-0.010 eV, PtNi: 0.046 eV). Moreover, calculations demonstrate that H₂O is preferentially adsorbed on Ni sites and the adsorption of OH on Ni sites is more stable than Nb sites. Ni sites on twin boundary of PtNiNb twin slab facilitate the H₂O dissociation with a highly reduced kinetics barrier compared with the Pt (111). Therefore, Nb doping greatly changes the electronic structure of PtNi/PtNiNb and Ni contributed to the water dissociation in the Volmer step.

The revisions corresponding to this comment are added in Page 7, Paragraph 1; Page 15, Paragraph 2; the “**Theoretical calculation on Pt-Ni-Nb Turing configurations**” section; Fig. 4, Fig. 5 and Supplementary Fig. 5 of the revised manuscript.

2. The authors have claimed many aspects regarding structural analyses. For example, 5-fold & 2-fold twins, stacking faults and lattice distortion on the PtNiNb sample. However, the modelling of DFT suggests Pt-Ni sample without Nb inside. What is the reason why the actual materials and the DFT model presents different structure? Please provide the DFT results for the actual model.

Answer: Thanks for the reviewer’s comments. Platinum and nickel are the major elements in the Turing PtNNb catalyst (Pt_{56.1}Ni_{33.5}Nb_{10.4}). In the previous manuscript, we intended to reveal the effects of nanotwins and lattice strain, which are the crucial characteristics of Turing structure, on catalytic activity. Therefore, niobium element and its effects on structure as well as catalytic properties were erroneously ignored. As for the structural aspect, DFT calculations are conducted on periodic structure. This precondition of DFT calculation limited the atomic model of multi-fold twins with intersected twin boundaries. However, the elementary unit of multi-fold twins is the twins with one boundary. And the sub-units of stacking faults highly resemble those of twin structure for an fcc structure. Therefore, this simplified PtNiNb twin model with

two parallel twin boundaries could be applied to reliably estimate the fundamental characteristic and properties of multi-fold twins and stacking faults.

According to the reviewer's advice, we have entirely re-constructed the calculation models based on the experimental composition of Turing PtNiNb ($\text{Pt}_{56.1}\text{Ni}_{33.5}\text{Nb}_{10.4}$). In the revised theoretical calculations, the atomic proportions of the fcc PtNiNb and fcc PtNiNb twin slab models are the same: $\text{Pt}_{55.6}\text{Ni}_{33.3}\text{Nb}_{11.1}$, close to the experimental composition of $\text{Pt}_{56.1}\text{Ni}_{33.5}\text{Nb}_{10.4}$. The subtle composition difference originates from the feature of atomic proportion in $\text{Pt}_{56.1}\text{Ni}_{33.5}\text{Nb}_{10.4}$, the same atomic composition is unavailable in the models with finite atoms. Hereafter, we re-conducted the theoretical calculations using the new atomic models, which aims to address the above concerns of the reviewer. **We also re-discussed the calculation results, and the following content is added to the revised manuscript as the previous "Theoretical calculation on Pt-Ni-Nb Turing configurations" section:**

The above analysis reveals Turing PtNiNb with the major crystallographic features of multiple nanotwins and lattice strain. We performed theoretical calculations to disclose the mechanism of enhanced HER activity induced by Turing structuring. Experimental data shows that the multi-fold nanotwins in Turing PtNiNb are mainly enclosed by $\{111\}$ and $\{200\}$ atomic planes with $[110]$ zone axis. We cleaved the fcc Pt_3Ni (110) layers and substituted Pt/Ni atoms to construct the four-atomic-layer fcc PtNi, fcc PtNiNb and fcc PtNiNb twin slab models. The fcc PtNi slab has a Pt : Ni ratio of 62.5 : 37.5, which is almost consistent with the composition of bimetal $\text{Pt}_{62.6}\text{Ni}_{37.4}$ sample. The atomic proportions of the fcc PtNiNb and fcc PtNiNb twin slab models are the same: $\text{Pt}_{55.6}\text{Ni}_{33.3}\text{Nb}_{11.1}$, close to the experimental composition of Turing PtNiNb ($\text{Pt}_{56.1}\text{Ni}_{33.5}\text{Nb}_{10.4}$). Extensive structure optimization calculations were carried out to determine the stable slab configurations, and the results of fcc PtNiNb twin slab model

is shown in Response-Figure 22a below.

The hydrogen adsorption Gibbs free energy (ΔG_{H^*}), a widely used descriptor for HER activity, was calculated to study the interaction between H^* and catalysts [J.K. Nørskov et al., J. Electrochem. Soc., 152 (2005), J23]. The Nb doping and the incorporation of twin-configuration in fcc PtNiNb slab model greatly change the local atomic configurations (Supplementary Fig. 23 and 24 of the revised manuscript). Response-Figure 22b shows the summary of calculated values of ΔG_{H^*} . The bimetal PtNi slab presents an optimal ΔG_{H^*} (0.046 eV, the bridge site of Ni-Ni) which is weaker than the ΔG_{H^*} of Pt (111) slab (-0.266 eV), revealing the positive contribution of Ni doping to catalytic activity. The Nb doping could greatly enrich the coordination types and optimize the ΔG_{H^*} of PtNiNb slab to -0.010 eV (the hollow site of Pt-2 shown in Supplementary Fig. 23 of the revised manuscript). Moreover, five other active sites of the PtNiNb slab also shows the ΔG_{H^*} values between ± 0.200 eV (the top site of Nb, the bridge sites of Pt-Nb and Ni-Ni, the hollow sites of Pt-1 and Ni, as shown in Supplementary Fig. 23 of the revised manuscript), suggesting the beneficial effects of Nb in improving the H^* adsorption/desorption on catalyst surface. As for the PtNiNb twin slab, calculation results show the optimized ΔG_{H^*} of 0.047 eV which is larger than the smallest ΔG_{H^*} of the PtNiNb slab. Despite of the inferior values of ΔG_{H^*} , the incorporation of twin configuration in PtNiNb twin slab has yielded more active sites compared with the PtNiNb slab. For example, nine active sites of PtNiNb twin slab show the ΔG_{H^*} values between ± 0.200 eV and four active sites have the ΔG_{H^*} values between ± 0.100 eV. Specifically, the atoms on and near the twin boundary (the top site

of Ni on TB; the top site of Nb near TB; the hollow site of Pt-2 on TB; as shown Supplementary Fig. 24 of the revised manuscript) contribute to the three of the most active sites. Twin configuration could result in partial symmetry breaking of a perfect fcc structure, and changes in orientation as well as interval of the atoms on sides of twin boundary. These crystalline features of twins are expected to induce significant changes in the local atomic coordination. The greatly increased number of active sites illustrate that the variation in atomic coordination induced by twin boundary can enormously improve catalytic activity. Hereafter, lattice strain was introduced in fcc PtNiNb slab. The statistical results on ΔG_{H^*} under compressive strain and tensile strain are shown in Response-Figure 22b. It is found that the compressive strain leads to the interaction optimization between H intermediates and catalyst surface. The PtNiNb twin exhibits the optimized ΔG_{H^*} values of -0.012 eV and about 0.000 eV under 1% and 2% compressive strain, respectively. This ΔG_{H^*} value (0.000 eV) of PtNiNb twin with 2% compressive strain (a hollow site of Pt on twin boundary) is close to the optimum value of ΔG_{H^*} for HER (0 eV), indicating the ideal moderate interaction between intermediates and active sites [J.K. Nørskov et al., *J. Electrochem. Soc.*, 152 (2005), J23]. According to Sabatier principle, this interaction, neither strong nor weak, is ideal for the adsorption/desorption of reactive participants [Z.W. She et al., *Science*, 355 (2017), eaad4998] This ideal interaction assures the desorption of H^* to form H_2 facilely in the Tafel step. The weaker adsorption of hydrogen on Turing PtNiNb was further confirmed by CV measurements with the left shift of underpotentially deposited hydrogen (H_{upd}) peak from 0.236 V vs. RHE for Pt/C to 0.198 V vs. RHE for Turing

PtNiNb (Supplementary Fig. 25 of the revised manuscript) [X. Li et al., Nat. Commun. 9 (2018), 4958]. Furthermore, the PtNiNb twin with 2% compressive strain shows a significantly increased density of active sites: eight active sites with ΔG_{H^*} values between ± 0.200 eV, which is 2-fold the PtNiNb twin without lattice strain. In this case, the atoms on the twin boundary contribute seven considerably active sites with the ΔG_{H^*} values between ± 0.100 eV (the top sites of Pt and Nb on TB; the hollow sites of Pt-1, Pt-2 and Pt-3 on TB; the hollow site of Pt-4 near TB; the bridge site of Ni-Ni-1 near TB; as shown in Supplementary Fig. 24 of the revised manuscript). These results illustrate that twin boundary could provide extra active sites. In contrast, tensile strain results in the general reduction in ΔG_{H^*} of PtNiNb twin. The reduction in ΔG_{H^*} suggests the enhanced adsorption of reaction intermediates on the catalyst surface, thus slowing the H^+ conversion and H_2 evolution.

The optimized interaction between reaction intermediates and catalyst surface may be attributed to the changes in electronic structure. The projected density of states (PDOS) of d orbital were further analysed and the results are shown in Response-Figure 22. Ni has a higher density near the Fermi level compared with Pt, and this trait indicates that Ni could donate more free electrons to near atoms during HER. Nb shows a broad band, but relative low density of occupation states. Nb doping in PtNi results in downshift of d -band centre from -2.82 eV to -2.94 eV for Pt sites (Response-Figure 22c and d). The downshift of d -band centre indicates that the transferred electronic would fill d -band, i.e., the antibonding electronic states [B. Hammer et al., *Nature* 376 (1995), 238] [J. K. Norskov et al., *Proc. Natl. Acad. Sci. U. S. A.* 108 (2011), 937]. Therefore,

Nb-doping could suppress the over-binding of H* on catalyst surface and weakening the H* adsorption, improving ΔG_{H^*} to optimal value. This result is in accordance with the optimized ΔG_{H^*} in comparison with PtNiNb (-0.010 eV) and PtNi (0.046 eV). The effects of twin configuration on electronic structure were also investigated. And the results show that the introduction of twin configuration in PtNiNb lead to a upshift of *d*-band center from -2.94 eV (PtNiNb without twin) to -2.86 eV (Response-Figure 22e). Such changes in *d*-band centre provide the theoretical interpretation to the ΔG_{H^*} deterioration from -0.010 eV (PtNiNb without twin) to 0.047 eV (PtNiNb twin). Nevertheless, the *d*-band center of the atoms on twin boundary was downshifted by 0.13 eV due to the introduction of twin configuration. The twin boundary may thus provide more active sites for the catalysis. Furthermore, the shift of *d*-band center can be tailored by lattice strain. As shown in Response-Figure 22f, the *d*-band centers of PtNiNb twin and Pt sites are monotonously shifted towards negative energy direction by increasing the compressive strain. These results are consistent with the findings of ΔG_{H^*} calculations.

In alkaline HER, proton generally originates from the dissociation of water and this reaction is always considered as the rate-determining step. The experimental results suggest that the HER on Turing PtNiNb surface follows the Volmer-Tafel pathway. The transition states (TS) were searched along the corresponding reaction pathway and the reaction barriers are calculated. We first calculated the possible adsorption configuration of H₂O and OH. It's found that H₂O is preferentially adsorbed on Ni and Nb sites, but OH could be more stably adsorbed on Ni sites. The initial state and final

state are therefore designed with H₂O-adsorbed on Ni sites, and OH-adsorbed on Ni sites with H-adsorbed on Pt sites, respectively. Both the PtNiNb and PtNiNb twin are calculated using the similar reaction pathway (Response-Figure 22h and i). The water dissociation on PtNiNb twin were proceeded on twin boundary due to the stable adsorption configuration. Calculation results demonstrated that the kinetic barriers of PtNiNb twin slab for H₂O dissociation (0.7042 eV) was lower than those of PtNiNi slab (0.791 eV) and Pt (111) slab (Response-Figure 22g). Besides, lattice strain facilitated water dissociation and further reduced the kinetic barrier to 0.605 eV. The lower dissociation barrier of water could markedly assist the proton production from water and facilitate the subsequent H₂ evolution. Therefore, twin configuration and lattice strain can significantly contribute to the high catalytic activity of PtNiNb in alkaline HER.

Response-Figure 22 **a**, Top view and side view of PtNiNb twin calculation model, where the grey grid represents twins configuration and red dashed lines are twin boundaries. **b**, The hydrogen adsorption Gibbs free energy (ΔG_{H^*}) of bimetal PtNi, PtNiNb and PtNiNb twin models, the synergic effects of twin configuration and lattice strain on ΔG_{H^*} are investigated. The projected electronic density of states for *d* orbital of PtNi (c), PtNiNb (d) and PtNiNb twin (e), including the PDOS diagram of free H₂O molecule. **f**, The *d*-band centres of PtNiNb twins as a function of tensile strains (positive) and compressive strains (negative). **g**, Free-energy diagram for water dissociation on Pt, PtNiNb, PtNiNb twins and PtNiNb twins with 2% compressive strain. The dynamic atom-configurations for water dissociation on the facet of PtNiNb (h) and PtNiNb twins with 2% strain (i), including the H₂O adsorption, the transition state and the final state.

3. If the sample can be prepared by a thin-film type (authors have claimed), why the sample had to be dropped-coated on the carbon paper support? Because the nanostructure of the Turing PtNiNb is the major novelty of the work, I guess the sample

film had to be studied by transferring the other (plate-like) substrate. Please provide if the drop-casted sample can present identical structural benefits compared to a thin-film type of sample.

Answer: Thanks for the reviewer's comments. Regarding to the preparation of Turing PtNiNb, superthin metal layer was first deposited on a hard supporting substrate by co-sputtering metallic targets (physical vapour deposition). This kind of metal film is greatly flat, and the composition as well as film thickness can be well controlled. The thickness of the as-deposited PtNiNb thin film was about 6 nm. This kind of superthin films was easily broken and wrinkled as they are exploited from the substrate. Therefore, a free-standing and large-area thin film with the intact surface is technically unavailable, and the as-deposited PtNiNb thin film can hardly be transferred on other (plate-like) substrate for electrocatalytic tests. Nevertheless, the thin-film type of samples can be prepared by depositing metal films on carbon cloth and flat nickel plate. We prepared three types of samples for comparison, including the catalyst inks of Turing PtNiNb (bimetal PtNi nanosheet, monometallic Pt 2D metal) dropped on carbon cloth, the Turing PtNiNb film (bimetal PtNi film, monometallic Pt film) deposited on carbon cloth and flat nickel plate. These PtNiNb, PtNi and Pt samples are prepared respectively using the same sputtering method. As shown in the LSV curves (Response-Figure 23c), the catalytic activity of monometallic Pt 2D metal dropped on carbon cloth is greatly higher than those of Pt films deposited on carbon cloth and flat nickel plate. The Turing PtNiNb samples (Response-Figure 23a) and bimetal PtNi (Response-Figure 23b) samples show the similar difference in catalytic performance: the catalytic performance of the nanosheet-typed samples is superior to the film-typed sampled deposited carbon cloth electrode and nickel electrode. In addition, we also analyse the catalytic activity of Turing PtNiNb, bimetal PtNi, Pt and NiNb catalysts, which are all directly deposited on flat nickel and microscopically smooth carbon cloth. The LSV curves, shown in Response-Figure 23d and e, clearly illustrate that the catalytic activity of thin-film typed Turing PtNiNb is greater than those of thin-film typed PtNi, Pt and NiNb. This result is in accord with the activity difference reflected by drop-casted samples (Fig. 4a of the

revised manuscript). Thus, we think that drop-casted samples could possess identical structural benefits compared to thin-film typed samples.

Moreover, the design of dropping-coated samples can be attributed to the following two reasons: first, the nanosheet-typed catalytic ink can greatly increase the electrochemical active surface area (Supplementary Table 6 of the revised manuscript); second, we intended to research the catalytic performance of Turing catalysts and application potential in the anion-exchange-membrane water electrolyser. Membrane electrode and carbon supporting materials, as the gas diffusion layer, are frequently assembled in the device. The catalyst ink of Turing PtNiNb nanosheet can be sprayed on carbon cloth and carbon paper with controllable Pt loading mass. Therefore, the Turing PtNiNb can be compared with the commercial Pt/C (powder) with regard to the activity and stability of anion-exchange-membrane water electrolyser.

Response-Figure 23 The LSV curves of **a**, Turing PtNiNb, **b**, PtNi 2D and **c**, Pt 2D metal prepared via different methods. LSV curves comparison for samples **d**, deposited on carbon cloth and **e**, deposited on Ni plate.

4. The authors claimed that the strain effect comes from the twins or defects can be beneficial for this Turing system. Then, where are the active sites for the reaction? The 5-fold twins or defective sites themselves cannot be active sites? What is the role of Ni and Nb in the electrochemical aspect?

Answer: Thanks for the reviewer's comments. The interpretation to this comment consists of the following two parts: (1) The role of Ni and Nb in the electrochemical aspect; (2) The identification active sites.

(1) The role of Ni and Nb in the electrochemical aspect

Answer: This question is almost the same with the comment 1 provided by the reviewer #3. We have performed extensive experiments and DFT calculations to analyse the effects of Ni and Nb on electrocatalytic activity. The detailed discussion could be found in the above response to comment 1 provided by the reviewer #3.

(2) The identification active sites

Answer: Crystallography analysis reveals that the introduction of twins configuration could result in partial symmetry breaking of a perfect fcc structure, and changes in orientation as well as interval of the atoms on sides of twin boundary. These crystalline features of twins-configuration are expected to induce significant changes in the local coordination environment of atoms on and near the twin boundary. DFT calculations show that the incorporation of twin configuration in PtNiNb twin slab yielded more active sites compared with the PtNiNb slab. For example, nine active sites of PtNiNb twin slab show the ΔG_{H^*} values between ± 0.200 eV and four active sites have the ΔG_{H^*} values between ± 0.100 eV. As a contrast, PtNiNb slab without twin has only six active sites with the ΔG_{H^*} values between ± 0.200 eV and one active site with the ΔG_{H^*} values between ± 0.100 eV (Response-Figure 24b). Specifically, the atoms on and near the twin boundary (the top site of Ni on TB; the top site of Nb near TB; the hollow site of Pt-2 on TB; as shown in Supplementary Fig. 24 of the revised manuscript) contribute to the three of the most active sites in the PtNiNb twin. Hereafter, lattice strain was introduced in fcc PtNiNb slab. It is found that the compression strain leads to the interaction optimization between H intermediates and catalyst surface. The PtNiNb twin exhibits the optimized ΔG_{H^*} values of -0.012 eV and about 0.000 eV under 1% and 2% compression strain, respectively. Furthermore, the PtNiNb twin with 2%

compressive strain shows a significantly increased density of active sites: eight active sites with ΔG_{H^*} values between ± 0.100 eV, which is 2-fold the PtNiNb twin without lattice strain. In this case, the atoms on the twin boundary contribute seven considerably active sites with the ΔG_{H^*} values between ± 0.100 eV (the top sites of Pt and Nb on TB; the hollow sites of Pt-1, Pt-2 and Pt-3 on TB; the hollow site of Pt-4 near TB; the bridge site of Ni-Ni-1 near TB; as shown in Supplementary Fig. 24). These results illustrate that twin boundary could provide extra active sites, especially the Pt sites on twin boundary. Furthermore, calculation results show that Ni on twin boundary are stable site for the adsorption of H_2O and OH. The significantly reduced kinetics barrier of H_2O dissociation on The Ni site on twin boundary, which could accelerate the H_2 evolution and greatly improve the catalytic activity (Response-Figure 24c ,d and e).

The revisions corresponding to this comment are added in the “Theoretical calculation on Pt-Ni-Nb Turing configurations” section (Page25 and Page26) of the revised manuscript.

Response-Figure 24 a, Top view and side view of PtNiNb twin calculation model, where the grey grid represents twins configuration and red dashed lines are twin boundaries. **b**, The hydrogen adsorption Gibbs free energy (ΔG_{H^*}) of bimetal PtNi, PtNiNb and PtNiNb twin models, the synergic effects of twin configuration and lattice

strain on ΔG_{H^*} is investigated. **c**, 加粗 Free-energy diagram for water dissociation on Pt, PtNiNb, PtNiNb twins and PtNiNb twins with 2% compressive strain. The dynamic atom-configurations for water dissociation on the facet of PtNiNb (d) and PtNiNb twins with 2% strain (e), including the H₂O adsorption, the transition state and the final state.

5. Is the sample controllable for the atomic composition? Since the authors claimed that the topological properties can be beneficial for the Turing catalyst system and provided DFT calculations to support them. More electrochemical/experimental evidence needs to be provided by different compositional PtNiNb Turing samples.

Answer: Thanks for the reviewer's comments. We carried out supplementary experiments to reveal the composition-structure relations of Pt-Ni-Nb metallic catalysts. Platinum is a widely studied active element for HER. Platinum content is speculated to be a significant contributor to the catalytic activity. Thus, we focus on the effects of platinum content on the structure of Pt-Ni-Nb catalysts. Meanwhile, the ratio of nickel to niobium is fixed at approximately 3:1 (the same as the Ni-Nb ratio in Turing Pt_{56.1}Ni_{33.5}Nb_{10.4}). These new nanocatalysts and Turing PtNiNb nanocatalyst were synthesized using the same method. The TEM images of these PtNiNb catalysts are shown in Response-Figure 25. Pt_{56.1}Ni_{33.5}Nb_{10.4} catalyst (denoted as Turing PtNiNb) presents the clear characteristics of Turing structure which is the same as previous findings (Response-Figure 25d). As the platinum content decreasing from 56.1 at.% to 51.7 at.%, the stripes tend to grow in-plane and parts of coarsening stripes interconnected to form large junctions (Response-Figure 25c). Furthermore, the traits of Turing structure eventually disappeared in Pt_{44.3}Ni_{42.1}Nb_{13.6} sample (Response-Figure 25a and b). However, higher platinum contents could also cause the breakdown of Turing structure. As shown in Response-Figure 25, Pt_{60.1}Ni_{30.8}Nb_{9.1}, Pt_{69.2}Ni_{23.6}Nb_{7.2}, Pt_{73.4}Ni_{20.1}Nb_{6.5} and Pt_{76.9}Ni_{17.8}Nb_{5.3} samples are planar nanosheets without the features of Turing structures. All the high-platinum-content samples (≥ 60.1 at.%) are composed of classical nanocrystals and do not have particular topographical characteristics. These

results suggest that the composition of Turing structure seems to be limited to a narrow range.

Subsequently, we tested the catalytic performance of all the as-prepared PtNiNb catalysts by dropping the corresponding catalyst ink on carbon cloth. The LSV curves are shown in Response-Figure 26. Although these Pt-Ni-Nb nanosheets (without Turing structure) has significantly different platinum content, Turing PtNiNb shows the highest catalytic activity among all the prepared Pt-Ni-Nb catalysts (Supplementary Fig. 21). For example, the overpotentials of Pt_{44.3}Ni_{42.1}Nb_{13.6} nanosheet and Pt_{76.9}Ni_{17.8}Nb_{5.3} nanosheets are 47.8 mV and 29.1 mV mV at 10 mA cm⁻², larger than that of the Turing PtNiNb (Pt_{56.1}Ni_{33.5}Nb_{10.4}, 18 mV). This finding demonstrate the structural advantages of Turing structure in electrocatalysis.

The revisions corresponding to this comment are added in Page 6, Paragraph 2; Page 21, Paragraph 2; Supplementary Fig. 4 and Supplementary Fig. 21 of the revised manuscript.

Response-Figure 25 a. The HRTEM image of Pt_{44.3}Ni_{42.1}Nb_{13.6} nanosheet. TEM images of Pt_{44.3}Ni_{42.1}Nb_{13.6} nanosheet (b), Pt_{51.7}Ni_{37.0}Nb_{11.3} nanosheet (c), Turing PtNiNb (d), Pt_{60.1}Ni_{30.8}Nb_{9.1} nanosheet (e), Pt_{69.2}Ni_{23.6}Nb_{7.2} nanosheet (f), Pt_{73.4}Ni_{20.1}Nb_{6.5} nanosheet (g), Pt_{76.9}Ni_{17.8}Nb_{5.3} nanosheet (h). i. The HRTEM image of Pt_{76.9}Ni_{17.8}Nb_{5.3} nanosheet.

Response-Figure 26 LSV curves of the Pt-Ni-Nb nanosheets with different compositions tested in 1.0 M KOH, with 85 % resistance compensation.

6. Please provide the acidic HER performance.

Answer: Thanks for the reviewer's comments. In the previous manuscript, we had actually clarified the HER performance of Turing PtNiNb in 0.5 M H₂SO₄ solution (Page 18, Paragraph 1 of the revised manuscript). The detailed results were illustrated in Response-Figure 27 of the revised manuscript. In the acidic HER case, Turing PtNiNb exhibited excellent catalytic activity with a mass activity 24.8 times higher than that of commercial Pt/C catalyst. The long-term stability of Turing PtNiNb was also demonstrated through galvanostatic measurement in 0.5 M H₂SO₄. These acidic HER results extend the generic significance of Turing structuring for electrocatalysis. Considering that this work focused on the alkaline HER performance of Turing PtNiNb, we made a concise comparison between Turing PtNiNb and Pt/C.

Response-Figure 27 a, LSV curves measured in $0.5 \text{ M H}_2\text{SO}_4$ of Turing PtNiNb and Pt/C. **b**, Mass activity measured in $0.5 \text{ M H}_2\text{SO}_4$ of Turing PtNiNb and Pt/C. **c**, Comparison for overpotential at 10 mA cm^{-2} and mass activity at 50 mV (vs. RHE) between Turing PtNiNb and Pt/C. **d**, Galvanostatic measurement in $0.5 \text{ M H}_2\text{SO}_4$ continuing 24 hours at 20, 50 and 100 mA cm^{-2} for Turing PtNiNb. The galvanostatic curves are without iR compensation.

REVIEWERS' COMMENTS

Reviewer #1 (Remarks to the Author):

The authors have revised the manuscript very well. This research is interesting and could attract wide attention from scientists in the field of materials and catalysis. Thus, I suggest the publication of it in Nature Communication in present form.

Reviewer #2 (Remarks to the Author):

The authors have made a commendable effort to address the concerns raised by all reviewers. The manuscript is greatly improved, especially in the aspect of doping effect, electronic structure analysis, and theoretical calculations. Firstly, the role of Ni and Nb atoms in the Turing PtNiNb structure was investigated from three parts: (1) structural aspect; (2) catalytic performance; (3) theoretical analysis. Secondly, the XAS was conducted to thoroughly analyze the atomic valence state and coordination structure. Thirdly, the theoretical models were rebuilt to take the doping elements and twin configuration into consideration. I believe the primary issues have been well resolved. Now, I recommend its publication after the authors address some minor issues:

1. As the authors claimed, the growth mechanism of Turing structure can be explained by reaction-diffusion theory with a pair of activator-inhibitor chemical substances. In the contrast experiments, the moderate Pt content and indispensable Nb dopant are confirmed. This work may be very intriguing if the compelling underlying mechanism can be well elucidated. Certainly, it does not affect the main conclusion of this paper.
2. The overpotential of commercial 20 wt% Pt/C catalyst in alkaline media should be much lower than the present data in the manuscript (51.0 mV).
3. As for the Pt L3-edge XANES spectra, the peak intensity is more commonly used to identify the atomic valence state (Nature 2022, 611, 284-288). The partially enlarged image of Figure 3a should be provided. For the Turing PtNiNb structure, is the positively charged entirety stable, considering that the Pt has a near-zero valence state and Ni/Nb owns positive valence state?

Reviewer #3 (Remarks to the Author):

The authors have addressed my comments well with sufficient additional results. The revised manuscript is acceptable for publication.

Reviewers' comments:

Reviewer #1 (Remarks to the Author): The authors have revised the manuscript very well. This research is interesting and could attract wide attention from scientists in the field of materials and catalysis. Thus, I suggest the publication of it in Nature Communication in present form.

Answer: Thanks for the reviewer's positive comments and recommendation.

Reviewer #2 (Remarks to the Author): The authors have made a commendable effort to address the concerns raised by all reviewers. The manuscript is greatly improved, especially in the aspect of doping effect, electronic structure analysis, and theoretical calculations. Firstly, the role of Ni and Nb atoms in the Turing PtNiNb structure was investigated from three parts: (1) structural aspect; (2) catalytic performance; (3) theoretical analysis. Secondly, the XAS was conducted to thoroughly analyze the atomic valence state and coordination structure. Thirdly, the theoretical models were rebuilt to take the doping elements and twin configuration into consideration. I believe the primary issues have been well resolved. Now, I recommend its publication after the authors address some minor issues:

1. As the authors claimed, the growth mechanism of Turing structure can be explained by reaction-diffusion theory with a pair of activator-inhibitor chemical substances. In the contrast experiments, the moderate Pt content and indispensable Nb dopant are confirmed. This work may be very intriguing if the compelling underlying mechanism can be well elucidated. Certainly, it does not affect the main conclusion of this paper.

Answer: Thanks for the reviewer's positive comments on these revisions.

2. The overpotential of commercial 20 wt% Pt/C catalyst in alkaline media should be much lower than the present data in the manuscript (51.0 mV).

Answer: Thanks for the reviewer's comments. Overpotentials of 20 wt.% Pt/C catalysts could be determined by Pt mass loading. According to the editor's advice in

the first revision process, we had reduced the Pt mass loading of the 20 wt.% Pt/C catalyst to $0.065 \text{ mg}_{\text{Pt}} \text{ cm}^{-2}$, the same with the Pt mass loading of the Turing PtNiNb sample for reasonable comparison. In order to address this comment, we conducted additional experiments to test a series of 20 wt.% Pt/C catalysts with different Pt mass loadings. As shown in Response-Figure 1 and Response-Table 1, the overpotential values of 20 wt.% Pt/C catalysts generally decrease with increasing Pt mass loading. The overpotentials of these samples are 51 mV ($0.065 \text{ mg}_{\text{Pt}} \text{ cm}^{-2}$), 45.9 mV ($0.098 \text{ mg}_{\text{Pt}} \text{ cm}^{-2}$), 40.0 mV ($0.13 \text{ mg}_{\text{Pt}} \text{ cm}^{-2}$) and 40.9 mV ($0.19 \text{ mg}_{\text{Pt}} \text{ cm}^{-2}$), respectively. These overpotentials of 20 wt.% Pt/C catalysts are in accordance with the data in the reported references (Response-Table 1). Therefore, we think that the presented data in this manuscript is reliable.

Response-Figure 1. LSV curves of Pt/C catalyst with different Pt mass loadings in 1.0 M KOH.

Response-Table 1. Overpotentials of 20 wt.% Pt/C catalysts tested in 1.0 M KOH and the corresponding Pt mass loading.

Mass loading (mg _{Pt} cm ⁻²)	Overpotential (mV)	Ref.
0.065	51.0	This work
0.0975	45.9	This work
0.13	40.0	This work
0.19	40.9	This work
0.1	78.0	L. Cao et al., Nature Catalysis 2 (2019), 134-141
0.2	49.0	B. Lu et al., Nature Communication 10 (2019), 631
0.2	40.0	J. Su et al., Nature Communication 8 (2017), 14969

3. As for the Pt L3-edge XANES spectra, the peak intensity is more commonly used to identify the atomic valence state (Nature 2022, 611, 284-288). The partially enlarged image of Figure 3a should be provided. For the Turing PtNiNb structure, is the positively charged entirety stable, considering that the Pt has a near-zero valence state and Ni/Nb owns positive valence state?

Answer: Thanks for the reviewer's suggestion. We have carefully read this reference proposed by the reviewer (Nature 2022, 611, 284-288) and cited it in the manuscript. As shown in the Response-Figure 2, we have added the enlarged image of Pt white line peaks in the XANES spectrum to clearly illustrate the changes in peak intensity. As for the valence state of Turing PtNiNb, the Ni and Nb elements are positively charged and the positive valence states may be attributed to the formation of Ni-oxides and Nb-oxides. Fitting calculations on the EXAFS spectra illustrate the Ni-O and Nb-O coordination (Supplementary Table 2 and Table 3 of the revised manuscript). However, the major coordination of Ni and Nb elements are still Ni-M (M = Ni and Pt) and Nb-M (M = Nb, Ni and Pt) coordination. These results are in accordance with the results of high-resolution X-ray photoemission spectra (Supplementary Figure 8 and Figure 20 of the revised manuscript). In addition, the stability of the Turing PtNiNb entirety was further confirmed by the results of X-ray photoemission spectra (Supplementary Figure 20 of the revised manuscript). Only

subtle changes in the elemental valences and peak positions of Turing PtNiNb could be detected after the long-term stability test (200 mA cm⁻², 60h).

Response-Figure 2. XANES spectra collected at Pt L₃-edge.

Reviewer #3 (Remarks to the Author): The authors have addressed my comments well with sufficient additional results. The revised manuscript is acceptable for publication.

Answer: Thank the reviewer for the recommendation to publish this work.